



# Bayesian uncertainty quantification in aerosol optical depth retrieval applied to TROPOMI measurements

Anu Kauppi[1,2], Antti Kukkurainen[3,4], Antti Lipponen[3], Marko Laine[1], Antti Arola[3],
Hannakaisa Lindqvist[1], and Johanna Tamminen[1]

[1]Finnish Meteorological Institute, Space and Earth Observation Centre, Helsinki, Finland
[2]Department of Mathematics and Statistics, University of Helsinki, Helsinki, Finland
[3]Finnish Meteorological Institute, Atmospheric Research Centre of Eastern Finland, Kuopio, Finland
[4]Department of Applied Physics, University of Eastern Finland, Kuopio, Finland

**Correspondence:** Anu Kauppi (anu.kauppi@fmi.fi)

**Abstract.**

We present here an aerosol model selection based statistical method in Bayesian framework for retrieving atmospheric aerosol optical depth (AOD) and pixel-level uncertainty. Especially, we focus on to provide more realistic uncertainty estimate by taking into account a model selection problem when searching for the solution by fitting look-up table (LUT) models to a satellite measured top-of-atmosphere reflectance. By means of Bayesian model averaging over the best-fitting aerosol models we take into account an aerosol model selection uncertainty and get also a shared inference about AOD. Moreover, we acknowledge model discrepancy, i.e. forward model error, arising from approximations and assumptions done in forward model simulations. We have estimated the model discrepancy empirically by a statistical approach utilizing residuals of model fits. We use the measurements from the TROPOspheric Monitoring Instrument (TROPOMI) onboard the Sentinel-5 Precursor in ultraviolet and visible bands, and in one wavelength band 675 nm in near-infrared, in order to study the functioning of the retrieval in a broad wavelength range.

We exploit a fundamental classification of the aerosol models as weakly absorbing, biomass burning and desert dust aerosols. For experimental purpose we have included some dust type of aerosols having non-spherical particle shapes. For this study we have created the aerosol model LUTs with radiative transfer simulations using the libRadtran software package. It is reasonably straightforward to experiment with different aerosol types and evaluate the most probable aerosol type by the model selection method.

We demonstrate the method in wildfire and dust events in a pixel level. In addition, we have evaluated in detail the results against ground-based remote sensing data from the AErosol RObotic NETwork (AERONET). Based on the case studies the method has ability to identify the appropriate aerosol types, but in some wildfire cases the AOD is overestimated compared to the AERONET result. The resulting uncertainty when accounting for the model selection problem and the imperfect forward modelling is higher compared to uncertainty when only measurement error is included in an observation model, as can be expected.



# 1 Introduction

Satellite aerosol retrieval algorithms are constantly improved to respond to the needs from wide and various user groups in the field of climate system research and air quality monitoring. The uncertainty analysis of retrieval process is essential in order to utilize the results of satellite retrievals properly. Quality flags can provide information about the potential failure of the retrieved pixel to the data user. A statistical reasoning is especially used when studying uncertainty originating from the methodological structure or from the assumption needed in the modelling of the physical system. For instance, Povey et al.

(2015) discuss the sources of uncertainty and uncertainty estimation in satellite remote sensing. The assessment of uncertainty estimates on the variety of satellite aerosol retrievals, also in the pixel level, are discussed by Sayer et al. (2020). On the other hand, there are studies to correct inaccuracies in the results of satellite aerosol product by model enforced post-processing using machine learning techniques as presented by Lipponen et al. (2021). The ongoing research with uncertainty analysis and improved uncertainty estimates benefits the data users as well as the retrieval algorithm developers.

In this study we use the measurements from the TROPOspheric Monitoring Instrument (TROPOMI) onboard the European Space Agency (ESA) Sentinel-5 Precursor (S5P). The TROPOMI observations are used for monitoring the Earth's atmosphere and climate system in order to provide data for applications e.g. for air quality services, aviation safety, and monitoring of CO and $NO_2$ emissions (Veefkind et al., 2012). The TROPOMI level 2 data include products of ultraviolet (UV) aerosol index, aerosol layer height, carbon monoxide, formaldehyde, methane, nitrogen dioxide, sulphur dioxide, ozone, surface UV, and

cloud properties. The operational level 2 TROPOMI UV aerosol index product developed by Royal Netherlands Meteorological Institute (KNMI) actually has two products, one for wavelength pair 340/380 nm and the other for 354/388 nm (Stein Zweers, 2018; Kooreman et al., 2020). The UV aerosol index product works also in the presence of clouds; thus, it is suitable for tracking the evolution of aerosol plumes from e.g. dust storms, wildfire, and volcanic ash eruptions. The other operational aerosol product developed by KNMI is the level 2 TROPOMI aerosol layer height product that provides the height of the

free-tropospheric aerosol layer for cloud-free scenes (Sun et al., 2019). The algorithm is based on a neural network method using absorption at the $O_2A$ band in the near-infrared wavelength range. Moreover, there is NASA's research aerosol algorithm TropOMAER that uses TROPOMI near-ultraviolet radiances to retrieve aerosol optical depth (AOD), single-scattering albedo, and UV aerosol index (UVAI) (Torres et al., 2020). The retrieval algorithm is based on algorithms which are the Ozone Monitoring Instrument (OMI) near-UV aerosol data product (OMAERUV; Torres et al. (2007, 2013, 2018)) for cloud-free

scenes and the OMACA retrieval for above-cloud aerosol scenes (Jethva et al., 2018). The recent paper Rao et al. (2021) presents a study using the synthetic and real TROPOMI measurements from the $O_2A$ band (758-771 nm) for retrieving the AOD and the aerosol layer height together utilizing the Bayesian approach. The retrieval algorithm is based on microphysical aerosol model selection and their results showed that when using the Bayesian averaging method the accuracy of the solution gets better in case of there is difficulty in the proper aerosol model selection.





In this paper we present a methodology for supporting the uncertainty analysis in the retrieval of AOD. The work is continuation of previous studies that were applied to the OMI measurements (Määttä et al., 2014; Kauppi et al., 2017). This time the methodology is applied for TROPOMI observations in the broad wavelength range including the ultraviolet and visible bands, and one wavelength band 675 nm in near-infrared. Basically, the retrieval method follows the OMI multi-wavelength aerosol data product (OMAERO; Torres et al. (2007); Curier et al. (2008)) in a sense that it uses spectral top-of-atmosphere (TOA) reflectances at several wavelength bands, and the AOD is retrieved based on the look-up tables (LUT) containing aerosol microphysical models. We utilize Bayesian inference when estimating the AOD at a reference wavelength 500 nm and its uncertainty originating from the aerosol model selection from the LUTs (MacKay, 1992; Spiegelhalter et al., 2002). We chose to use as a retrieved point value for AOD the maximum a posterior (MAP) estimate. In addition, we consider modelling error originating from imperfect forward modelling and call this additional error term a model error or a model discrepancy (Kennedy et al., 2001; Brynjarsdóttir et al., 2014). For this study, we simulated TROPOMI measurements with radiative transfer calculations using the libRadtran software package (Mayer et al., 2005; Emde et al., 2016) in order to create the multi-dimensional aerosol model LUTs. As a base of LUTs structure we followed the OMI LUTs description in the OMAERO algorithm (Torres et al., 2007; Veihelmann et al., 2007). We have included some dust models having a non-spherical particle shape representing new type of models in this context.

In Sect. 2 we describe the used data products and databases in the retrieval. In Sect. 3 we discuss construction of the LUTs regarding radiative transfer calculations as well as selection of the aerosol types and aerosol properties. Section 4 gives the description of the statistical retrieval methodology and approach to uncertainty quantification. Finally, in Sect. 5 we demonstrate the methodology with case studies and the results are evaluated using aerosol data provided by the AErosol RObotic NETwork (AERONET) that is a federation of ground-based remote sensing aerosol networks (Holben et al., 1998). The summary of this study and conclusions are given in Sect. 6.

## 2 Description of the used data

### 2.1 TROPOMI/S5P data

The TROPOMI instrument is onboard the S5P satellite launched on 13 October 2017. The S5P is part of the EU Copernicus program and is a collaborative mission by the ESA and the Netherlands. The S5P is a polar-orbiting satellite on a sun-synchronous orbit with an equatorial crossing time at ∼13:30 solar time. TROPOMI is a nadir-viewing imaging spectrometer having a 2600 km wide swath. It has a good spatial resolution as the ground pixel size at nadir can be as small as 5.6 x 3.5 km², however the resolution depends on the data product. TROPOMI measures in the ultraviolet, visible, near-infrared and shortwave infrared wavelength range between 270 and 2358 nm.

Further details about the TROPOMI data used in this study can be found in Appendix A. For this retrieval study, we selected 16 wavelength bands altogether from the TROPOMI UV-visible (UVIS) detector spectral bands 3 and 4, and from the near infrared (NIR) band 5. We have used about 1 nm-wide wavelength bands centered at 342.5, 354.0, 367.0, 376.5, 388.0 and 399.5 nm (from band 3); 406.0, 416.0, 425.5, 436.5, 440.0, 451.5, 463.0, 483.5 and 494.5 nm (from band 4), and 675.0 nm



(from band 5). We retrieve AOD at the reference wavelength 500 nm. We like to note that we have not carried out a thorough investigation about the best choice of TROPOMI wavelength bands for this type of multi-wavelength aerosol retrieval. We have included most of the same wavelength bands as the OMI multi-wavelength algorithm covers (Torres et al., 2007; Veihelmann et al., 2007). Additionally, the bands 440 nm and 675 nm were selected since the AERONET provides AOD also at these wavelengths. In Fig. 1, the TROPOMI reflectance spectrum computed from the level 1b radiances and irradiance from the orbit 09989 is shown as an example for one pixel located in northern Africa. The selected wavelengths from the detector bands 3 and 4 (upper panel) and from the band 5 (bottom panel) are marked by vertical lines.

We did the following quality checks to the TROPOMI ground pixels used in the example cases (see Sect. 5). First, we used quality confirmation information given by 'ground_pixel_quality' variable from the level 1b band 3 radiance datafile and omitted the pixel if that value differed from zero (Rozemeier et al., 2019). Moreover, we required for the ground pixel to have solar zenith angle (SZA) maximum of 75°. Additionally, we checked the pixel quality by utilizing a quality flag variable 'qa_value' from the S5P level 2 UV aerosol index product. In our case we rejected the pixel if 'qa_value' was below 0.5 (Apituley et al., 2021). We did not perform an actual cloud screening since we considered only limited sets of pixels so far.

## 2.2 Surface reflectance

Assumption of a surface reflection is a significant source of uncertainty in the satellite aerosol retrieval algorithms. The reflectance from the surface varies according to the surface soil type and is mixing with the TOA backscattered solar radiance. Especially, the separation of the atmospheric backscattered reflectance from the reflection of bright surfaces, such as desert dust areas or snow cover, is challenging. In this study we have used the surface reflectance data from the Surface reflectance DAtabase for ESAs earth observation Missions (ADAM) (https://earth.esa.int/eogateway/catalog/adam-surface-reflectance-database-v4-0, last access: 9 September 2021). The ADAM product has a global map of monthly mean climatologies of the surface reflection at 0.1° spatial resolution for land and for ocean. We have used the API toolkit provided by ADAM portal to calculate a collection of multidirectional reflectances for the selected viewing geometries over land surfaces at our choice of wavelength bands. The main reason for choosing ADAM database was its ability to provide the surface reflectance data according to instrument's measuring geometry, i.e. viewing zenith angle, relative azimuth angle and SZA. The ADAM products are available for the registered users via the ADAM portal.

## 3 Definition of aerosol model properties and creation of the look-up tables

We have pre-calculated the multidimensional LUTs for different TROPOMI sun-satellite geometries and for various aerosol properties presented in Sect. 3.1. The aerosol properties are given as input to the radiative transfer run introduced in Sect. 3.2. Thus the aerosol properties included determine the different aerosol types stored in the LUTs.



## 3.1 Aerosol types

We have classified the aerosol types by their origin and optical properties as weakly absorbing (WA), biomass burning (BB) and desert dust (DD) aerosols. These main aerosol types are split into subtypes so that we have a total of 66 aerosol models

(i.e. LUT models). We have made use of the OMI OMAERO LUTs (Torres et al., 2007; Veihelmann et al., 2007; Curier et al., 2008) when assigning aerosol size distribution and refractive index for weakly absorbing, biomass burning and the main part of the dust aerosol models (see Table 1). The rest of the aerosol properties for dust type are collected from literature as listed in Tables 2 and 3. The variety of aerosol microphysical properties included is dedicated to this study and not intended to be a comprehensive collection of aerosol mixtures.

The aerosol subtypes are classified by the physical and optical properties of the particle ensembles. We have defined an identification (ID) number for each LUT model following the numbering in the OMAERO LUTs. The ID number has four digits so that the first digit symbolizes the main type, the second digit stands for the imaginary part of the complex refractive index, the third digit is for the vertical distribution and the fourth digit is for the size distribution. The intention of the model numbering is to distinguish the subtypes within the associated main type. The imaginary part of the refractive index indicates

the strength of absorption of the particles. The real part of the refractive index affects the scattering properties of the particles and has here a constant value that depends on the aerosol type. The BB and DD type of aerosol models have also been split into three categories according to the aerosol vertical layer distributed uniformly at 0-2 km, 2-4 km or 4-6 km. As a special case we have desert dust models '33xx' and '34xx' (see Table 2) where only the second digit separates these models according to the shape of particles; spherical ('3') and non-spherical ('4').

### 3.1.1 Weakly absorbing and biomass burning aerosol models

Weakly absorbing (WA) aerosol types consist of a mixture of urban, industrial and rural aerosols. Biomass burning (BB) type of aerosols in major part originate from natural sources such as wildfires. The chemical composition of the BB type of aerosols can consist of black carbon or organic carbon. It is also worth mentioning that vehicle emissions and combustion processes in urban or industrial surroundings can be a source of soot. We have the collection of properties for altogether nine WA models

and 27 BB models based on the OMAERO aerosol LUTs presented in Table 1. The size distribution is given by a bimodal log-normal function.

### 3.1.2 Dust models

There are a range of different types of dust with dissimilar mineralogy from arid and semi-arid regions; e.g., Sahara, Asia, Sahel, Middle East, Australia and South America. The dust particles have different mineralogical compositions and therefore

different scattering and absorption properties depending on their origin. We have included in the LUTs altogether 30 desert dust (DD) models. The DD models are listed separately in Tables 1, 2 and 3 according to the data source of the used aerosol properties i.e. the size distribution and the refractive index. The imaginary part of the refractive index is wavelength dependent and written down for each dust model in Table 4.





Even thought we have a good variety of DD models based on the OMAERO LUTs (Table 1), we found it interesting to add some new dust models including also particles with a non-spherical shape (Table 2). There are altogether 12 models listed in Table 2 with two different size distribution and two different shape of particles. We included six dust models which used a spheroidal particle shape instead of spherical. For simplicity, we used prolate spheroids with aspect ratio as 0.25 according to Wagner et al. (2012). In reality the dust particles have a variety of different shapes and are not perfect spheroids either (e.g. Wagner et al. (2012); Lindqvist et al. (2014); Huang et al. (2020)). We have taken the size distribution and the imaginary part of the complex refractive index from the paper Wagner et al. (2012) using data from sample Cairo 2 collected from the Northern Sahara. We have interpolated the imaginary part of the complex refractive index values to our used wavelengths. As seen in Tables 2 and 4, the models '33xx' and '34xx' have the same size distributions and refractive index values, except that the particle shape is different.

The third group of dust aerosol models (see Table 3) have the size distribution and complex refractive index values adapted from the aerosol properties used for climate simulations with the aerosol-climate model ECHAM5.5-HAM2 (Zhang et al., 2012; Räisänen et al., 2013). We included these models in our collection of LUTs for interest in order to expand the variety of dust types.

## 3.2 Forward model simulations

The forward model simulates the synthetic TOA reflectance as observed by the satellite instrument. The forward model simulation is carried out by a radiative transfer (RT) model for given atmosphere description and instrument related parameters e.g. wavelength range and measuring geometry. Here, in order to speed up the computational time in the retrieval process the RT simulations have been pre-computed with the RT model using various atmospheric states and for a range of different aerosol properties and loadings, and the results are stored in the multi-dimensional LUTs (see Table 5). Consequently, in the retrieval algorithm the solution is achieved by searching for the TOA spectral reflectance interpolated from the LUT that fits to the satellite measured spectral reflectance.

We have computed the RT simulations using the RT software package libRadtran (version 2.0.3) (Mayer et al., 2005; Emde et al., 2016) with the DISORT 2.0 and MYSTIC solvers for the radiative transfer. The key input data to the libRadtran are wavelength range, surface pressure, atmospheric model and aerosol optical properties. Moreover, the input data include observation geometry. We have used the U.S. 1976 standard atmosphere for atmospheric vertical profile. As output from libRadtran run we obtain atmospheric reflectance, transmittance and spherical albedo (see Table 6).

The aerosol microphysical properties (e.g. aerosol size distribution, particle shape and complex refractive index) have been converted to the optical properties using the Modelled Optical Properties of enSeMbles of Aerosol Particles (MOPSMAP) tool. The set of our used microphysical input variables are listed for different types of aerosols in Tables 1, 2 and 3. The MOPSMAP tool comprises a Fortran program for performing calculation and a web interface for interactive usage. In principle, the optical properties for user-given aerosol particle properties are calculated by interpolating on a MOPSMAP grid of particle sizes, shapes and refractive indices. More information and instructions to use the MOPSMAP tool are given in the user guide at https://mopsmap.net/mopsmap_userguide.pdf (last access: 9 September 2021) and in the article (Gasteiger et al., 2018). As



output we get the optical properties including phase matrix, single scattering albedo and Ångström exponent that are given as input to the libRadtran. Finally, the instrument geometry, atmospheric reflectance, transmittance, spherical albedo, and aerosol microphysical and optical properties are gathered together for each aerosol model and stored in corresponding LUT.

## 4 Methodology description

We retrieve AOD at the reference wavelength $\lambda_{\mathrm{ref}} = 500\,\mathrm{nm}$, denoted by symbol $\tau = \tau(\lambda_{\mathrm{ref}})$, and related uncertainty using statistical methodology based on the Bayesian inference (see e.g. Gelman et al. (2013)). In this approach the unknown AOD is handled as a random variable. The resulting AOD is expressed as a posterior density that provides retrieval uncertainty as well. The point value of the AOD is chosen to be the maximum a posterior (MAP) estimate that is the mode of the posterior curve. The same methodology has been applied earlier in the research study using the OMI measurements (Määttä et al., 2014; Kauppi et al., 2017). The AOD retrieval scheme is actually analogous to the algorithm used by the OMI multi-wavelength algorithm OMAERO (Curier et al., 2008) in a sense that it searches for the most appropriate aerosol LUT models exploiting the least squares fitting to the measured TOA spectral reflectance at several wavelength bands $\lambda$.

The aerosol models (i.e. LUTs) are discrete characterizations of the real atmosphere with different aerosol loadings. In addition, the models have been constructed with forward model calculations using assumptions and approximations of reality (see Sect. 3.2). In this work, we acknowledge this uncertainty originating from imperfect forward modelling. We have estimated the model discrepancy (i.e. model error) by making use of residuals of model fits as described in Sect. 4.1. We use a symbol $\eta(\lambda)$ for the model discrepancy and add it into the observation model

$$\boldsymbol{R}_{\mathrm{obs}}(\lambda) = \boldsymbol{R}_{\mathrm{mod}}(\tau, \lambda) + \eta(\lambda) + \epsilon_{\mathrm{obs}}(\lambda), \tag{1}$$

where $\boldsymbol{R}_{\mathrm{obs}}(\lambda)$ is the measured spectral TOA reflectance and $\boldsymbol{R}_{\mathrm{mod}}(\tau, \lambda)$ is the model-based reflectance. The measurement error due to instrument noise is assumed independent and zero mean Gaussian $\epsilon_{\mathrm{obs}}(\lambda) \sim N(0, \boldsymbol{\sigma}_{obs}^2(\lambda))$. For the sake of simplicity and since this is a research type of retrieval, we have used a spectrally constant value for a signal-to-noise ratio (SNR) of the instrument and set $\boldsymbol{\sigma}_{\mathrm{obs}}(\lambda) = \boldsymbol{R}_{\mathrm{obs}}(\lambda)/\mathrm{SNR}$. In the case studies presented in Sect. 5 we used the value SNR = 700 as a rough estimate.

We consider in this study pixels over land only. We use the following formula valid above a Lambertian surface (e.g. Chandrasekhar (1960)) to calculate the LUT model-based TOA reflectance

$$\boldsymbol{R}_{\mathrm{mod}}(\lambda, \tau, \mu, \mu_0, \Delta\phi, p_{\mathrm{s}}) = R_a(\lambda, \tau, \mu, \mu_0, \Delta\phi, p_{\mathrm{s}}) + \frac{A_{\mathrm{s}}(\lambda)}{1 - A_{\mathrm{s}}(\lambda)\,s(\lambda, \tau, p_{\mathrm{s}})} T(\lambda, \tau, \mu, \mu_0, p_{\mathrm{s}}), \tag{2}$$

where $A_{\mathrm{s}}$ is the surface reflectance, $R_a$ is path reflectance, $T$ transmittance and $s$ spherical albedo of the atmosphere as seen from below. The values of $R_a$, $T$ and $s$ are taken from the LUT (see Table 6) by interpolating within node points of the associated parameters $\tau$ (AOD), $\mu$ (cosine of viewing zenith angle), $\mu_0$ (cosine of solar zenith angle), $\Delta\phi$ (relative azimuth angle) and $p_{\mathrm{s}}$ (surface pressure). We assumed the Lambertian surface when simulating the LUT's reflectances with RT model. However, when implementing Eq. (2) in the example cases we have used instrument's viewing direction dependent surface





reflectivity $A_\mathrm{s}$ from the ADAM database (see Sect. 2.2). Nevertheless, the difficulty in satellite aerosol retrieval is that both the
215 surface and aerosols have direction dependent reflectance.

## 4.1 Estimation of Model discrepancy

Our intention is to get more realistic uncertainty estimates by acknowledging the model discrepancy when choosing the most
appropriate LUTs. We search for systematic differences between the modeled and observed reflectance that stands for model
errors due to forward model approximations. Therefore we formulate the difference between the observed physical system and
220 simulator output (i.e. LUTs) as discrepancy function.

The characterization of the model discrepancy is done in the same manner as in earlier studies when the OMI data was
used (Määttä et al., 2014) e.g. by utilizing a Gaussian process approach (Rasmussen and Williams, 2006). Here we give only a
brief description of the approach. We have proposed a zero mean Gaussian process model $\eta(\lambda) \sim \mathrm{GP}(0, C)$ where a covariance
function $C$ is obtained by statistical approach (Banerjee et al., 2004). We defined $C$ by characterizing wavelength dependent
systematic structure in the residuals of model fits $\boldsymbol{R}_\mathrm{res}(\lambda) = \boldsymbol{R}_\mathrm{obs}(\lambda) - \boldsymbol{R}_\mathrm{mod}(\tau, \lambda)$ i.e. by inspecting spatial (or spectral)
correlation of the residuals. In spatial statistics the dependence is described by a (semi)variogram function. First, we calculated
the empirical semivariance $\gamma(d)$ at wavelength pair distances $d = |\lambda_i - \lambda_j|$ as

$$\gamma(d) = \frac{1}{2} \frac{1}{n(d)} \sum_{d=|\lambda_i - \lambda_j|}^{n(d)} \left( \boldsymbol{R}_\mathrm{res}(\lambda_i) - \boldsymbol{R}_\mathrm{res}(\lambda_j) \right)^2, \tag{3}$$

where $n(d)$ is the number of wavelength pairs with the same distance $d$. We excluded the NIR wavelength band 675.0 nm
since it has such a long distance to the other wavelength bands in UV and VIS (see Fig. 1) that the spectral correlation is
not presumed. Next, we fitted a theoretical parametric semivariogram model from the literature (Banerjee et al., 2004) to the
empirical semivariogram and ended up to use a Gaussian variogram model. At the same time as fitting the theoretical Gaussian
variogram model to the empirical model we searched values for tuning parameters $l$, $\sigma_0^2$ and $\sigma_1^2$ of the theoretical model. A
correlation length $l$ describes the wavelength distance where the residuals are still correlated. In addition, a parameter $\sigma_0^2$ is
235 responsible for non-spectral diagonal variance and a parameter $\sigma_1^2$ for spectral variance. We ended up to use the parameter
values as $l=90$, and for both $\sigma_0^2$ and $\sigma_1^2$ values of $1\%$ of the measured reflectance. Finally, the values for the tuning parameters
$l$, $\sigma_0^2$ and $\sigma_1^2$ were used to derive the Gaussian process covariance function $C$ as

$$C(\lambda_i, \lambda_j) = \begin{cases} \sigma_1^2 \exp\left(-(\lambda_i - \lambda_j)^2/l^2\right), & \lambda_i \neq \lambda_j \\ \sigma_0^2 + \sigma_1^2, & \lambda_i = \lambda_j \end{cases}. \tag{4}$$

Finally, the covariance function $C$ forms the model error covariance matrix $\boldsymbol{C}$ that defines the allowed smooth departure for
the modelled reflectance from the observed reflectance.

Practically, the residuals of model fits $\boldsymbol{R}_\mathrm{res}(\lambda)$ were calculated for the pixel sets collected from two separate days, 24 July
2019 and 22 March 2019. The reflectance $\boldsymbol{R}_\mathrm{mod}(\tau, \lambda)$ is from the model that has the best match with the measurement in
the least squares sense. When doing residual analysis we assumed higher random noise level than used in the actual retrieval
and chose SNR = 300. Then the requirement for fit between the measured and modelled reflectance is loosened and thus the





number of successful retrievals increases. The pixel sets are globally distributed as were gathered from seven orbits crossing all the continents. There were total of 2779 pixels and the ratio of successful retrieval was $\sim 39\%$ for 24 July and $\sim 12\%$ for 22 March respectively. The ensemble of residuals involved can be considered as a representative sample since there were only 15 models out of 66 LUT models that never were selected as the best model. The models not selected were total of 11 of DD type and four of BB type. For interest, the BB models not selected ('2131', '2132', '2232', '2233') have aerosol height assigned to

be 4-6 km as expressed by the third digit of the model ID number. The DD models not being selected have no common feature except for major part of the models have aerosol vertical distribution at 2-4 km or 4-6 km.

## 4.2 Bayesian approach for AOD retrieval

We will present only the main idea here as the methodology based on the Bayesian inference is explained thoroughly in the papers (Määttä et al., 2014; Kauppi et al., 2017) when applied to the OMI measurements. As a priori assumption all the

255 LUT models are equally likely, that is we have not pre-selected models for different areas according to a land cover type or climatology for instance.

First, we search for the solution to $\tau$ (i.e. AOD at 500 nm) within each aerosol model LUT by fitting the modelled reflectance to the measured reflectance. By the Bayes formula the posterior distribution for $\tau$ is

$$p(\tau|\boldsymbol{R}_{\mathrm{obs}}, m) = \frac{p(\boldsymbol{R}_{\mathrm{obs}}|\tau, m)\,p(\tau|m)}{p(\boldsymbol{R}_{\mathrm{obs}}|m)} \tag{5}$$

assuming that the aerosol model $m$ is the correct one. We presume that a prior distribution $p(\tau|m)$ for $\tau$ within the model $m$ follows log-normal distribution (O'Neill et al., 2000) and as a consequence the AOD can get only positive values. The likelihood function when including the model error covariance matrix $\boldsymbol{C}$ is

$$p(\boldsymbol{R}_{\mathrm{obs}}|\tau, m) \propto \exp\left(-\frac{1}{2}\,\boldsymbol{R}_{\mathrm{res}}(\lambda)^{T}\left(\boldsymbol{C} + \mathrm{diag}\left(\boldsymbol{\sigma}_{\mathrm{obs}}^{2}(\lambda)\right)\right)^{-1}\boldsymbol{R}_{\mathrm{res}}(\lambda)\right). \tag{6}$$

The normalizing constant $p(\boldsymbol{R}_{\mathrm{obs}}|m)$ of posterior does not depend on $\tau$ and was calculated numerically in our case (see Määttä

et al. (2014) for details).

The competing aerosol models are compared using the probability value $p(\boldsymbol{R}_{\mathrm{obs}}|m_i)$ that is the probability of observing $\boldsymbol{R}_{\mathrm{obs}}$ given the model $m_i$, and we call this probability as an evidence value for the model $m_i$ in this context. We have chosen to select maximum of ten best models according to the model evidence values until a cumulative sum of the values exceeds the value of 0.8. Then we calculate the relative evidence for each selected model $m_i$ with respect to all the maximum of ten

selected models as

$$p(m_i|\boldsymbol{R}_{\mathrm{obs}}) = \frac{p(\boldsymbol{R}_{\mathrm{obs}}|m_i)}{\sum\limits_{j} p(\boldsymbol{R}_{\mathrm{obs}}|m_j)}, \tag{7}$$

where the denominator has the sum over the evidence values of the selected models. As a consequence we can infer that the higher relative evidence the more plausible the aerosol model is to explain the measurement.





Using the Bayesian model averaging (Hoeting et al., 1999) we can combine the posterior distributions (Eq. (5)) of $\tau$ for the selected models so that the individual posterior is weighted by its model's relative evidence (Eq. (7)) as

$$p_{\mathrm{avg}}\left(\tau|\boldsymbol{R}_{\mathrm{obs}}\right) = \sum_{i=1}^{n} p\left(\tau|\boldsymbol{R}_{\mathrm{obs}}, m_i\right) p\left(m_i|\boldsymbol{R}_{\mathrm{obs}}\right), \tag{8}$$

where $n$ is the number of selected models and has upper limit as ten in our application. The averaged posterior distribution (Eq. (8)) also reflects on uncertainty in model selection especially in case when the distinct LUT models give different AOD estimates. In that case the uncertainty incorporated in the averaged posterior may be larger than any single model has. The retrieved point value for AOD is chosen to be the MAP estimate, i.e. mode of the averaged posterior. Alternatively, the retrieved AOD estimate based on several appropriate models could be calculated as a sum of the MAP AOD estimates of selected models weighted by their relative evidences.

Finally, we do a goodness of fit test and check for the retrieved pixel that the selected model with the highest evidence has adequate fit to the measurement. We calculate a modified chi-squared function as

$$\chi^2 = \tfrac{1}{n-1}\boldsymbol{R}_{\mathrm{res}}(\lambda)^{\mathrm{T}}\left(\mathbf{C} + \mathrm{diag}(\boldsymbol{\sigma}_{\mathrm{obs}}^2(\lambda))\right)^{-1}\boldsymbol{R}_{\mathrm{res}}(\lambda), \tag{9}$$

where $n$ is the number of wavelength bands and $\mathbf{C}$ is the model error covariance matrix. We have set here the acceptance criterion as $\chi^2 \leq 2$. The main reasons for a rejected solution for the pixel is cloud contamination or cases where none of the models has good enough fit to the measurement indicating that the proper model might be missing from the ensemble of LUT models.

## 5 Case studies and results

Here we demonstrate the functioning of the methodology in several example cases. As results are the most probable aerosol models from the collection of LUTs and the corresponding MAP AOD estimate from the averaged posterior probability density. As stated above we consider only the over-land pixels.

As a particular example, Fig. 2 shows for three individual pixels that even though the different types of aerosol models have almost equally good fit with the measured reflectance (upper row), the MAP AOD estimates from the individual aerosol models can have a notable variation (bottom row). For instance, the first pixel (left) has the range of the AOD values with uncertainty between 0.18-0.34. This signifies the uncertainty in the retrieved AOD as it depends on the selection of the proper aerosol model. The averaged posterior distribution (Eq. (8)), plotted by black dashed curve, reflects the uncertainty in model selection.

In our study the retrieved point value for AOD is set to be the mode of the averaged posterior distribution, i.e. the MAP AOD estimate, indicated by a red dashed vertical line (Fig. 2, bottom row). The alternative point measure for the retrieved AOD that we like to show as an additional information is the sum of the relative evidence weighted MAP estimates of the individual models indicated by a black dashed vertical line (Fig. 2, bottom row). The MAP AOD estimate (red) and the weighted sum MAP AOD (black) may differ, as seen in Fig. 2 (middle panel). The relative evidence (%) (Eq. (7)) is reported next to each model ID number in the posterior density plots. It indicates how plausible that model is among all the selected best-fitting




models to explain the measurement. The posterior curves are colored differently according to main type, i.e. weakly absorbing type in blue, biomass burning type in red and desert dust type in brown (not shown in this case).

The results of the case studies are evaluated using aerosol data provided by the ERONET. We downloaded the Version 3 direct-sun Level 1.5, or Level 2.0 if available, quality-assured and cloud-screened AOD data for the selected AERONET sites. In addition, we obtain aerosol-type information, e.g. the size distribution and the refractive index, from the AERONET

Version 3 inversion product (e.g. Shin et al. (2019); Dubovik et al. (2000)). The AERONET sites involved and the collocated TROPOMI pixels are listed in Table 7. In this study we used a simple geometric collocation criteria where we selected a single TROPOMI ground pixel that has successful solution and which centre coordinates are closest to the AERONET site coordinates. A temporal collocation criteria is to use the AERONET AOD(500 nm) values that have been measured within a one-hour time window coinciding with the TROPOMI overpass time.

## 5.1 Case California fires 2020

In autumn 2020 there were severe and disastrous wildfires in the Southern coast of California. Here we consider two set of pixels on 23 August 2020 and demonstrate which aerosol models are selected as the most probable models and the corresponding AOD levels. The two pixel sets are roughly marked with red rectangles in the true-colour image of MODIS onboard the Aqua satellite (Fig. 3) provided by NASA's Earth Observing System Data and Information System (EOSDIS) Worldview tool

(https://worldview.earthdata.nasa.gov). The Aqua/MODIS has the equator crossing time about the same time as the TROPOMI has.

Figure 4 presents the spatial distribution of the retrieved MAP AOD estimate from the averaged posterior for the both pixel sets I (left) and II (right). The AOD values are high in the coast area (set I) whereas there is a considerable variation in the AOD values in the pixel set II. There are missing pixels where the retrieval failed due to not passing the quality checks, none of the

models fit to the measurement or failure of the goodness of fit test (Eq. (9)). The locations of the AERONET sites NASA_Ames (I), Monterey (I) and Univ_of_Nevada-Reno (II) are marked in the plots. Since the solution is missing at the location of the site NASA_Ames (I) we present the results for the pixel next to it in more detail (see Table 7 and Fig. 8).

The left panel in Fig. 5 shows the number of selected (i.e. the best-fitting) models for each pixel in the both pixel sets. We have selected maximum of ten models according to the evidence value that signifies how plausible the model is to explain

the measurements (see Sect. 4.2). As seen there are many pixels where single model is enough to describe the measurement. The right panel in Fig. 5 presents the distribution of the dominant aerosol type determined by the single best model that has the highest evidence. The colour scale for the aerosol types are: WA (blue), BB (red) and DD (brown). We can observe that biomass burning is the prevailing type in the pixel set I (Ib) as might be expected in this case.

In addition, we summed up pixel-by-pixel the relative evidences (%) of the selected models within each main type and call

it as a shared evidence (see Figs. 6 and 7 b-d). It expresses quantity of confidence for each main type. The shared evidence also informs if the solution for the pixel gives indication of one or several prevailing aerosol main types. Furthermode, the figure (a) shows the relative evidence (%) of the single best aerosol model having the highest evidence indicating how much this single





model explains the measurement with respect to the other selected models if any. These figures confirm that BB is the main type of the selected models in the pixel set I (Fig. 6), whereas both WA and BB type of models dominate in the set II (Fig. 7).

Figure 8 shows the results in more detail for the three TROPOMI pixels which have geometric collocation with the AERONET sites NASA_Ames (upper row), Monterey (middle) and Univ_of_Nevada-Reno (bottom row). The filled posterior density curves (right column) stand for the results when the model error was not taken into account (b and f). This result is missing for the pixel (d) since each candidate model has too high disagreement to the measured reflectance. We can clearly notice that we get higher AOD uncertainty, i.e. a width of the posterior density function is broader, when the model error

is included as compared to the results assuming no model error. When the model error is not acknowledged in the retrieval process we get presumably too optimistic uncertainty estimate. This is because we can require close fit between the modeled and observed reflectance when assuming that the forward model simulated LUTs represent real atmosphere. We can also note the difference in the retrieved AOD estimates (indicated by red vertical dashed lines) with a relative difference of the retrieved MAP AOD estimates being approximately 2.6 % for the first pixel (b) and 8.5 % for the third pixel (f) respectively.

The retrieved AOD is high (i.e values over 2) for the two pixels located in the coast area (Fig. 8b, d) and in both cases there is only one model that has sufficient fit to the measured reflectance. There are variety of BB and WA type of aerosol models selected for the third pixel (bottom row) and they represent also aerosols in the upper layer of the atmosphere as can be detected from the third digit in the model ID number. The distribution of AOD(500) values from NASA_Ames and Univ_of_Nevada-Reno measured within a one-hour time window including the TROPOMI overpass time are shown in Fig. 8 (b, f) indicated by

the vertical gray lines. The black vertical line points to the average of these AERONET AOD(500) values. We can conclude that in these two cases the retrieved MAP AOD estimates are comparable with the mean of AERONET AOD values (see Table 7). Our retrieval highly overestimates AOD ($\sim 3.0$) when compared to the AOD values measured at the Monterey site ($\sim 1.6$) (see Table 7). In this case the temporal collocation was poor since the last measurements of that day in the Monterey site were done about 2 hours before the TROPOMI overpass time. The mean of Ångström exponent 440–870 nm value of $> 1.6$ (not shown

here) in all these three AERONET sites considered reveal that there were fine mode particles present. Additionally, the spectral pattern of the imaginary index provided by the AERONET version 3 inversion product (not shown here) indicates significant presence of absorbing organic carbon (i.e. brown carbon). We like to note here that we do not contain in our collection of LUTs biomass burning aerosol model that has wavelength dependent imaginary part of the complex refractive index. Jethva et al. (2011) discuss the overestimation of AOD at 500 nm if absorption is assumed to be independent of the wavelengths.

### 5.2 Case South America: transported smoke from Australian bush fires in January 2020


The bushfires in Australia were disastrous in extent and damage continuing about nine months in distinct parts of Australia during season 2019-2020. Especially, from November 2019 to January 2020 the fire situation was critical until the rainfall in February helped to take control over fires. In the example day on 6 January 2020, the aerosol plume has been transported over the Pacific ocean, as can be seen in the EOSDIS Worldview plot of combined true-colour image of Visible Infrared

Imaging Radiometer Suite (VIIRS) and aerosol index from the Ozone Mapping and Profiler Suite (OMPS) onboard the joint NASA/NOAA Suomi NPP satellite (Fig. 9, left). This figure is provided by the Worldview tool as part of the story that presents





a satellite imagery related to the bushfires in Australia (https://worldview.earthdata.nasa.gov). The higher amount of absorbing aerosol particles can also be discovered in the figure of the TROPOMI UV aerosol index product from a wavelength pair 340/380 nm (Fig. 9, right). In this case study we examined how the transported smoke aerosols from Australian bushfires can
be detected in South America.

We can observe from Fig. 10 that in general terms the MAP AOD estimate based on the single model having the highest evidence (a) produces higher AOD values than when the MAP AOD estimate is taken from the averaged posterior (b) that combines all the selected best-fitting models. The location of the AERONET site Pilar_Cordoba is marked with a black circle. We get high AOD levels (i.e. values over 4) for the pixels vicinity of the Pilar_Cordoba site. Overall, the range of the retrieved
AOD values is wide. There are only one or two aerosol models selected for the majority of the pixels, as seen in Fig. 11a and Fig. 12a. The prevailing main type is biomass burning (see Fig. 11b and Fig. 12c). There are also some pixels with the mixture of BB and DD types as can be seen from Fig. 12c, d.

Figure 13 shows the results for two TROPOMI pixels in the vicinity of the AERONET site Pilar_Cordoba. The pixel collocated with the site Pilar_Cordoba (b) has much higher MAP AOD estimate ($\sim 4.3$) than measured by the ground-based
instrument ($\sim 0.7$) (see Table 7). The other pixel (d), located a little bit further, has as a result lower AOD ($\sim 2$). As discussed above (see Sect. 5.1) the reason for retrieving very high AOD estimate in this case could be a lack of proper aerosol model of biomass burning type. The site Pilar_Cordoba has the Ångström exponent 440–870 nm value of $\sim 1.1$ (not shown here) around the TROPOMI overpass time denoting that there were fine mode particles present. It is noteworthy that the selected models, 'BB2323' and 'BB2233', assume aerosols to be positioned at higher altitude at 2-4 km and 4-6 km respectively (see Sect. 3.1).
This suggests a realistic functioning of the aerosol model selection as the transported smoke plume presumably appears at higher layer in the atmosphere (e.g. Gonzales et al. (2020)).

## 5.3   Desert dust cases

This case study considers detecting dust aerosols originating from the Sahara on two example days. The first case, on 21 February 2021, consists of a set of pixels from the orbit 17409 around the AERONET site Medenine_IRA in Tunisia near the
source of dust. The other day, 24 February 2021, considers dust aerosols transported from the Sahara to Europe in the pixel set from the orbit 17452 around the AERONET site Bure_OPE in France. In Fig. 14 are shown the true-colour Aqua/MODIS images from the EOSDIS Worldview tool for the both example days.

We report first the results for the day 21 February 2021. The AERONET site Medenine_IRA is marked with a black circle in the plot of the MAP AOD estimates (Fig. 15). Since the retrieval failed for that pixel, the more detailed results are shown in
Fig. 18 for the pixel close to it (marked with black rectangle). As can be seen in Fig. 15, Fig. 16b and Fig. 17 the aerosol type is WA or BB for the pixels with enhanced AOD nearby the site Medenine_IRA. Otherwise the prevailing aerosol type is desert dust and there are pixels where several models of DD type contribute to the retrieved AOD (see Fig. 16 and Fig. 17).

Figure 18 shows results in more detail for three pixels where the uppermost pixel (first row) has the closest location to the AERONET site Medenine_IRA. As can be seen (left column) the selected modelled reflectance (green curve) for the pixel (a)
does not agree so well with the observation (blue curve). We can also notice that the solution is missing for that pixel (b) when





the model error was not taken into account. The reason is that the candidate models had too high difference to the observed reflectance and thus the retrieval failed. But, when the model error is acknowledged it is allowed a smooth departure from the modelled to the observed reflectance determined by the model discrepancy term (see Eq. (1)). As seen in Fig. 18b, the ground-based mean value AOD(500) ∼ 1.209 is comparable with the MAP AOD estimate ∼ 1.16 and is within the uncertainty

limits of the retrieved AOD. Also the AERONET data for the site Medenine_IRA indicate presence of coarse particles since the mean value of Ångström exponent (440–870 nm) is ∼ 0.091 (not shown here). For the pixel, shown in the middle row, there is a number of DD type of models selected which can explain the measurement. We can also notice for the pixels (d) and (f) that the selected models represent both type of particle shapes included, i.e. spherical and non-spherical. The selected models assume also different altitudes of the aerosol layers. Thus we could infer from the results for the two pixels (middle

and bottom) that there are a mixture of dust aerosols distributed in the different altitudes. However, we should also consider the possible effects of surface reflectance in this case.

The other example day 24 February 2021 considers pixel set in Europe. We can see from Fig. 19 that there are enhanced AOD levels in the vicinity of the AERONET site Bure_OPE that is marked by a black circle. There are actually two TROPOMI pixels that cover the location of the Bure_OPE site and the resulting aerosol type for these two pixels are of BB and DD type

respectively (see Fig. 20b and 21). Overall, when the selected type is DD then the AOD level is higher (see Figs. 19, 20 and 21).

In Fig. 22 are shown detailed results for the two pixels collocated with the site Bure_OPE. The both pixels have as a result the same MAP AOD estimate (∼ 0.53) that is rather well comparable with the mean AOD value (∼ 0.45) from the site Bure_OPE (see Table 7). The pixel in the upper row has two BB type of models ('BB2311' and 'BB2312') selected as the most probable models and they differentiate with size distribution (see Table 1). There is no result when the model error is not acknowledged.

The pixel below (Fig. 22c, d) has as a result two DD type of models ('DD3311' and 'DD3411') which differentiate only by the particle shape assumption (see Table 2). Especially, the best-fitting model is 'DD3411' (non-spherical shape) when the model error was not acknowledged (filled posterior in (d)). As seen in Fig. 22d the models with different shape assumptions give separate AOD posterior densities and therefore the resulting averaged posterior (black dotted curve) has clearly two peaks. However, in this case the model 'DD3311' has higher relative evidence (63.4 %) and thus the averaged posterior has

higher peak for that model. In Fig. 22d we can also note a small difference between the MAP AOD estimate (∼ 0.53) and the evidence weighted sum of the MAP AOD estimates of the selected models (∼ 0.58) (see Sect. 4.2). The AERONET data at the site Bure_OPE reveal that there were coarse particles present since the reported daily mean value of Ångström for 24 February 2021 is 0.193 whereas the monthly mean for February 2021 is 1.017.

## 6    Discussion and conclusions

This paper presents the methodology for uncertainty analysis in the LUT-based satellite aerosol retrieval when applied to the TROPOMI/S5P measurements. The retrieved quantity is the AOD jointly with the most probable aerosol types. In this study we have focused on to take into account uncertainty originating from approximations in the forward modelling and to handle the model selection problem when choosing the most probable aerosol models from the LUTs.





We have utilized the Bayesian inference when searching for the AOD at 500 nm with uncertainty expressed by posterior
probability distribution. The presented aerosol retrieval is performed in two steps; first we calculate the AOD posterior distri-
bution and the evidence value for each aerosol model and in the second step we compare these models and select the most
probable models (e.g. maximum of ten) according to the evidence values. The selection of one single best model is not always
meaningful as there can be several models that can contribute to explain the measurement. The uncertainty due to difficulty
in the aerosol model selection is resolved by the Bayesian model averaging approach. By model averaging we retrieve the
averaged posterior and can incorporate uncertainty in the model selection into the result. The point AOD value reported is
chosen to be the MAP AOD estimate from the averaged posterior.

The model discrepancy signifies non-modelled systematic differences between the observed and modelled reflectance. When
using measurements from the broad wavelength range it gives reason to examine the correlated error structure between the
neighbouring wavelengths. When acknowledging the model discrepancy (i.e. the model error) as the additional error term in
the observation model, we allow the modelled reflectance to have smooth deviation from the measurement and consequently
get higher uncertainty in the resulting AOD than if not including the model error. We attempt to get more reliable uncertainty
estimates when accounting for that the LUTs are discrete approximations of the reality.

For this study we made the forward model simulations in order to create the LUTs of the aerosol optical and microphysical
properties composed of weakly absorbing, biomass burning and desert dust types. Consequently, the candidate aerosol types
are fully determined by the aerosol properties we have given as input to the radiative transfer run. The accuracy of the retrieval
is depending on the correctness of the aerosol optical and microphysical properties considered. As a special feature in this study
we have included aerosol models of dust type with non-spherical shape of particles. The particle shape can have a large effect
on the scattering properties. It is expected that the aerosol properties included do not cover all the possible aerosol scenarios.
For instance, biomass burning type of aerosols having spectrally dependent absorption and atmospheric ageing would be worth
of studying. In addition, for global analysis a sea salt and volcanic aerosol types should be included. We like to acknowledge the
importance of laboratory chamber measurements and field campaigns to collect the absorbing and scattering aerosol properties.
Measuring and analyzing both the size distribution and refractive index together, or most preferably the scattering matrices, are
valuable reference data for the radiative transfer calculations when constructing the LUTs. For instance in the paper Di Biagio
et al. (2019) is presented new dataset for mineral dust aerosols based on soil samples from different desert areas.

In order to check that the solution is acceptable we determined the goodness of fit threshold value to inspect if the model
with the highest evidence has appropriate fit to the measurement. Especially, we encountered for some pixels (not shown here)
an obvious difference between the measured and modelled reflectance at wavelength centered at 675 nm from the TROPOMI
band 5. The reason can be the cloud contamination, the effect of surface reflectance or that the proper aerosol model is missing.
This could be subject for the further investigation. Also, subject for the further study could be investigating if the NIR band
675 nm does give additional information with respect to the other wavelength bands included.

We have tested the retrieval algorithm performance in the wildfire case and also its ability to detect the transported smoke
aerosols from long distance over the Pacific ocean. The detection of a variety of desert dust types in Africa and transported dust
aerosols in Europe is also examined. The test cases show that the retrieval can obtain the expected aerosol types, but the AOD





level is not always comparable with the AERONET AOD(500 nm). For instance, the reason for notable overestimation in the
biomass burning example cases could be an absence of the proper aerosol LUT models (e.g. Jethva et al. (2011)). Interestingly,
the dust models with non-spherical particle shape were selected as the best-matching models for some pixels which agrees
with the general consensus on using spheroidal models for the interpretation of remote sensing observations for dust aerosols
(e.g. Dubovik et al. (2006); Merikallio et al. (2011)). We need to do more retrieval exercises and verify the results e.g. with the
ground-based AERONET data before we can make conclusions about the retrieval accuracy. Particularly, statistical comparison
studies using e.g. monthly data from the AERONET or the satellite aerosol products (e.g. VIIRS), would give more information
about functionality of the presented method.

We see a potential for this kind of retrieval algorithm that attempts to quantify the uncertainty estimates in conjunction with
retrieving the aerosol type and the AOD level. For instance, the presented methodology could benefit the following studies:

1. Experiment with new LUT models with different aerosol microphysical and optical properties, e.g. BB type of models
   with spectrally dependent absorption or atmospheric ageing or both, and examine the results. Analyze also the optical
   properties, e.g. scattering phase matrix, of the selected aerosol models.

2. Analyze how the retrieval method using the LUTs can detect and classify different aerosol situations and atmospheric
   ageing (e.g. aerosols near the source, transported aerosols or mixed aerosol events), and potentially get information about
   aerosol layer altitude.

3. Study what kind of connections can be found in the results between nearby pixels.

4. Explore the effect of different surface reflectance assumptions on the model selection and the resulting AOD.

5. Investigate the possibility to use the presented methodology in producing training data or in learning process for algo-
   rithms utilizing machine learning.

6. Compare the results with the other satellite aerosol products (e.g. TROPOMI/S5P and VIIRS).

*Data availability.* The TROPOMI level 1b and level 2 UV aerosol index product data used in this study are available for download at the
Sentinel-5P Pre-Operations Data Hub (https://s5phub.copernicus.eu). The results of the case studies are available from the corresponding
author on request.

## Appendix A: Using TROPOMI/S5P data

We have used the radiance and irradiance data from the TROPOMI level 1b version 1 product of the UV-visible (UVIS) and
near infrared (NIR) bands. We downloaded the TROPOMI level 1b and level 2 data from the Sentinel-5P Pre-Operations Data
Hub (https://s5phub.copernicus.eu). We selected the level 1b irradiance product that has the sensing time closest to the sensing



time of the level 1b radiance product. The latitude and longitude values, assigned to the ground pixel centre, were taken from the level 1b band 3 radiance datafile.

*Author contributions.* AKa, ML and JT contributed to generation of the statistical methodology used in this study. AKu performed the forward model simulations. AKa, AKu, AL, AA and HL contributed to collection and discussion of the aerosol properties included. AL gathered the surface reflectance data set from the ADAM portal. AKa performed the case studies. AKa wrote the major parts of the text. All the authors contributed to editing the manuscript.

*Competing interests.* The authors declare that they have no conflict of interest.

*Acknowledgements.* This work has been funded by the Finnish Centre of Excellence in Inverse Modelling and Imaging granted by the Academy of Finland (336798). We appreciate the Copernicus Open Access Hub for providing TROPOMI/Sentinel-5P level 1b and level 2 data. We acknowledge by thanks the whole TROPOMI/S5P team. We thank the AERONET PIs and Co-Is and their staff for establishing and maintaining the 6 sites used in this investigation. The ADAM products have been collected from the ADAM database (adam.noveltis.com), and were produced by the ADAM team under the European Space Agency (ESA) study contract Nr C4000102979. We acknowledge the use of imagery from the NASA Worldview application (https://worldview.earthdata.nasa.gov), part of the NASA Earth Observing System Data and Information System (EOSDIS). Special thanks to our colleagues at FMI; Päivi Haapanala for encouraging discussion about radiative attenuation by aerosols, Pekka Kolmonen for kind help with wavelength band issues and Petri Räisänen for friendly help providing the dust aerosol microphysical properties applied with ECHAM5 model.






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





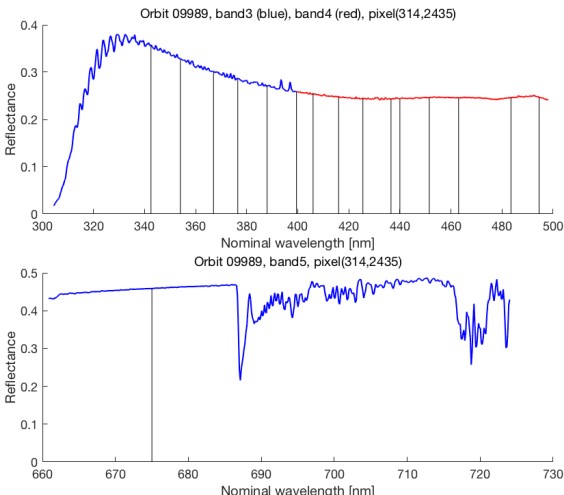

**Figure 1.** The TROPOMI reflectance spectrum from Bands 3 (in blue) and 4 (in red) (upper row) and from Band 5 (bottom row) for one pixel from the orbit 09989 on 17 September 2019. The vertical lines indicate the selected 16 wavelength bands.

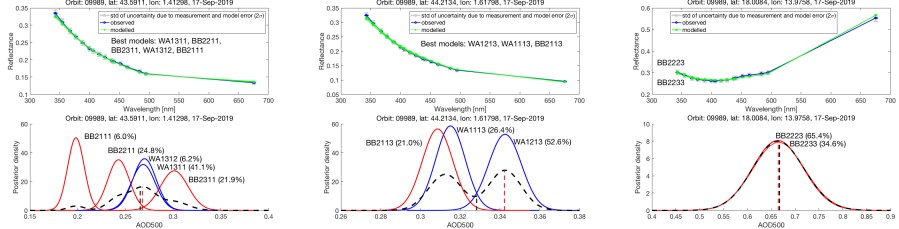

**Figure 2.** Three individual pixels from the orbit 09989 on 17 September 2019. The observed reflectance (blue dots) and the modelled reflectances (green dots) of the best-fitting models at the selected wavelength bands are shown in upper row. The error bars in grey correspond to 2 x standard measurement and forward model error. The posterior probability distributions of the AOD at 500 nm for the best-fitting models are shown in bottom row. The black dashed curve is the averaged posterior distribution taken over the best models. The red dashed vertical line indicates the MAP AOD estimate from the averaged posterior distribution. The black dashed vertical line indicates the AOD value that is the sum of the evidence weighted MAP estimates of the individual models.

Zhang, K., ODonnell, D., Kazil, J., Stier, P., Kinne, S., Lohmann, U., Ferrachat, S., Croft, B., Quaas, J., Wan, H., Rast, S., and Feichter, J. The global aerosol-climate model ECHAM-HAM, version 2: sensitivity to improvements in process representations. Atmos. Chem. Phys. 12: 8911–8949. https://doi.org/10.5194/acp-12-8911-2012, 2012.






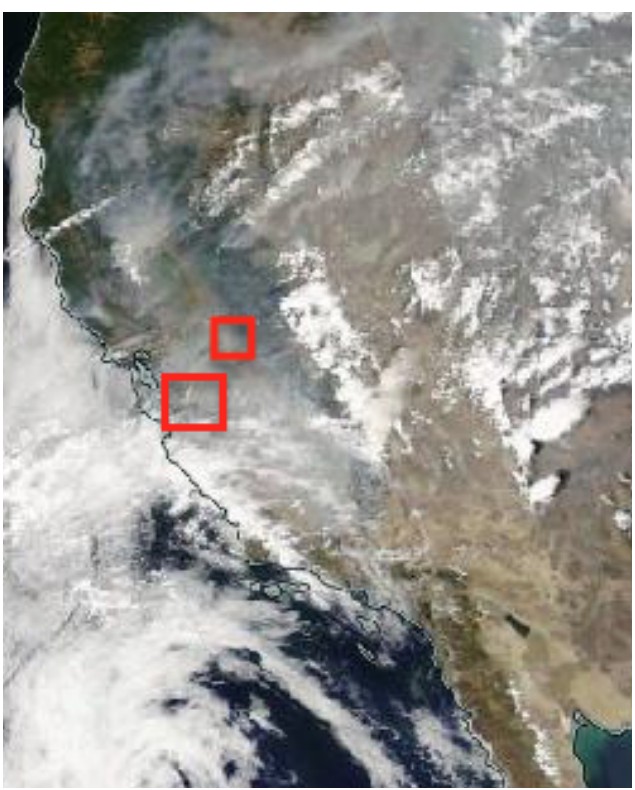

**Figure 3.** True-colour Aqua/MODIS image from the EOSDIS Worldview tool (https://worldview.earthdata.nasa.gov) showing smoke over the Western California coast area on 23 August 2020. The two areas examined are marked with red rectangles.

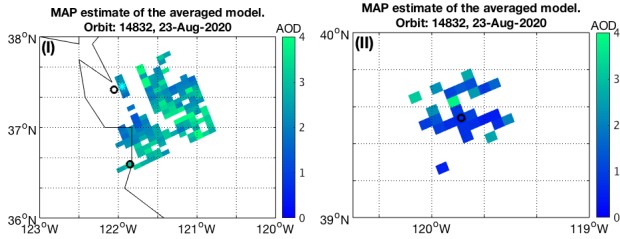

**Figure 4.** On 23 August 2020, orbit: 14832. **(I, II)** The spatial distribution of the retrieved MAP AOD estimate in the two separate pixel sets examined. The locations of the AERONET sites NASA_Ames **(I)**, Monterey **(I)** and Univ_of_Nevada-Reno **(II)** are marked by black circles. Since the solution is missing for the pixel at NASA_Ames **(I)** we present in Fig. 8 the results in detail for the pixel next to it marked here by a light blue star (see Table 7).



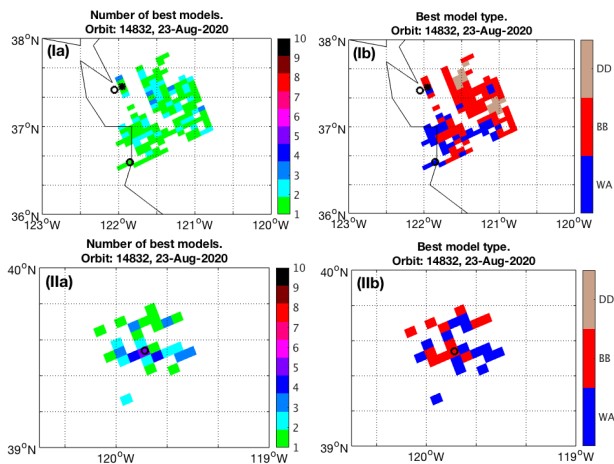

**Figure 5.** On 23 August 2020, orbit: 14832. **(Ia, IIa)** The number of the selected best models for the two pixel sets considered. **(Ib, IIb)** The spatial distribution of the main aerosol type of the best model with the highest evidence. The locations of the AERONET sites NASA_Ames and Monterey **(Ia, Ib)** and Univ_of_Nevada-Reno **(IIa, IIb)** are marked by black circles. The retrieval pixel (50,2605) next to NASA_Ames is marked by a black star.

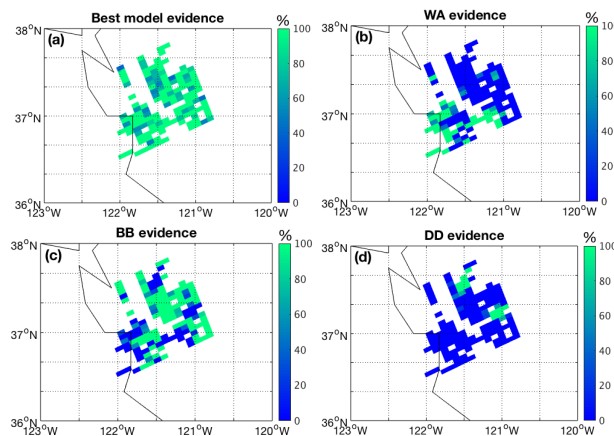

**Figure 6.** On 23 August 2020, orbit: 14832. The spatial distribution of the relative evidence (%) for the retrieved pixels of set I. **(a)** The relative evidence (%) of the single best model having the highest evidence. The shared evidence (%) of all the best models within each main aerosol type: **(b)** weakly absorbing, **(c)** biomass burning and **(d)** desert dust.





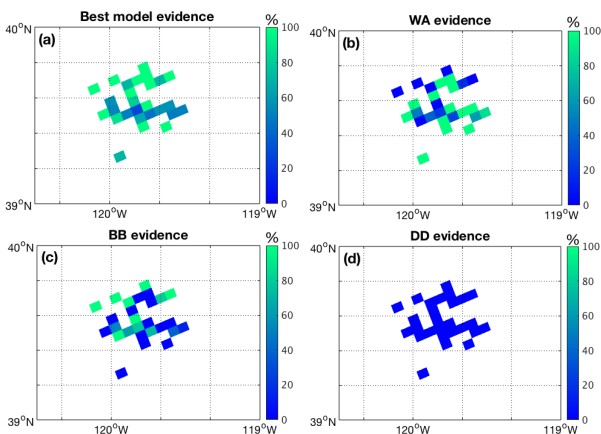

**Figure 7.** On 23 August 2020, orbit: 14832. Same as Fig. 6, but showing the spatial distribution of the relative evidence (%) for the retrieved pixels of set II.



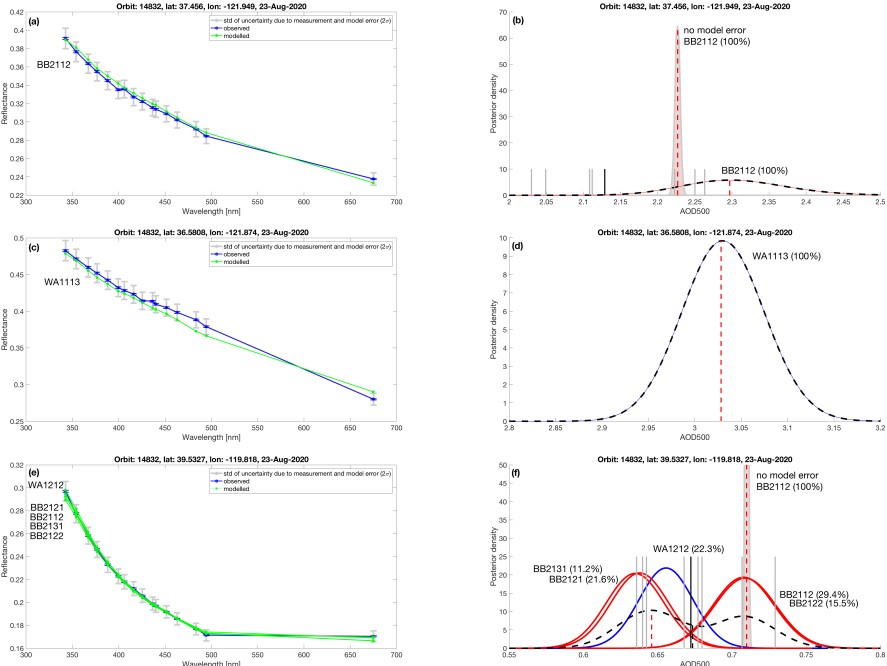

**Figure 8.** On 23 August 2020, orbit: 14832. Results shown for the three TROPOMI ground pixels (i.e. (50,2605), (47,2589) and (84,2627)) collocated to AERONET sites listed in Table 7: **(a, b)** NASA_Ames , **(c, d)** Monterey and **(e, f)** Univ_of_Nevada-Reno. The observed reflectance (blue dots) and modelled reflectances (green dots) of the best-fittings models with the error bars in grey corresponding to 2 x standard measurement and forward model error (left column). The posterior probability distributions of AOD for each best-fitting model (right column). The black dashed curve is the averaged posterior distribution taken over the best models. The filled posterior curve represents the result when model discrepancy is not included **(b, f)**. The red dashed vertical line indicates the MAP AOD estimate from the averaged posterior distribution. The gray vertical lines in **(b)** and **(f)** denote the AERONET AOD(500) values within a one-hour time window including the TROPOMI overpass time and the black solid vertical line is the average value of them.

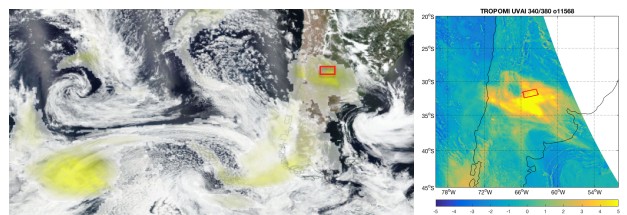

**Figure 9.** On 6 January 2020 smoke aerosols transported from Australia to South America as seen in the EOSDIS Worldview combined plot of Suomi NPP/VIIRS reflectance (true-colour) and Suomi NPP/OMPS aerosol index (left). The aerosol index from the TROPOMI UV aerosol index product from the 340/380 nm pair is shown in the right hand side. The area examined is marked with a red rectangle.





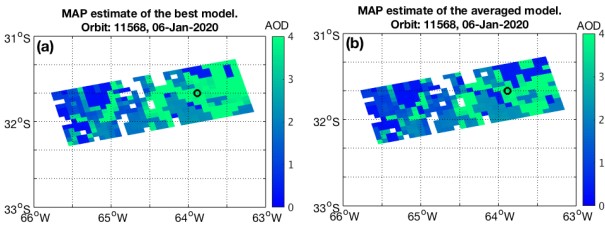

**Figure 10.** On 6 January 2020, orbit: 11568. **(a)** The MAP AOD estimate from the single best model with the highest evidence. **(b)** The MAP AOD estimate from the averaged posterior distribution of the selected best models. The location of AERONET site Pilar_Cordoba is marked by a black circle.

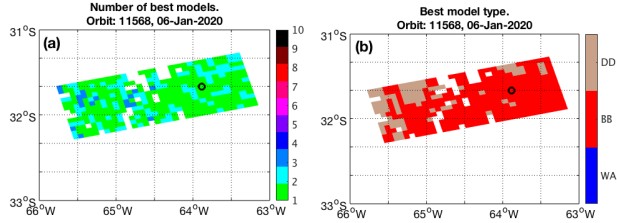

**Figure 11.** On 6 January 2020, orbit: 11568. **(a)** The spatial distribution of the number of the selected best models and **(b)** the main aerosol type of the single best model with the highest evidence. The location of AERONET site Pilar_Cordoba is marked by a black circle.

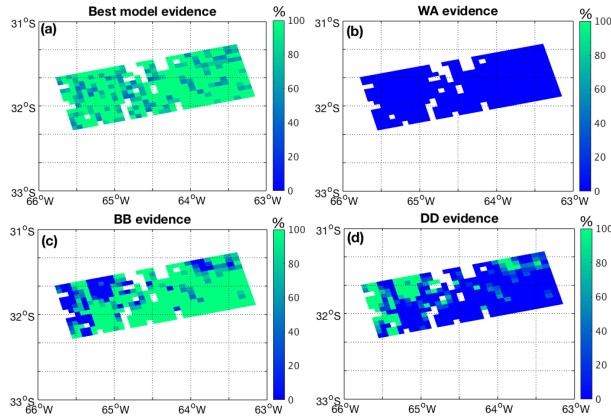

**Figure 12.** On 6 January 2020, orbit: 11568. The spatial distribution of the relative evidence (%). **(a)** The relative evidence (%) of the single best model having the highest evidence. **(b-d)** The shared evidence (%) of all the best models within each main aerosol type.



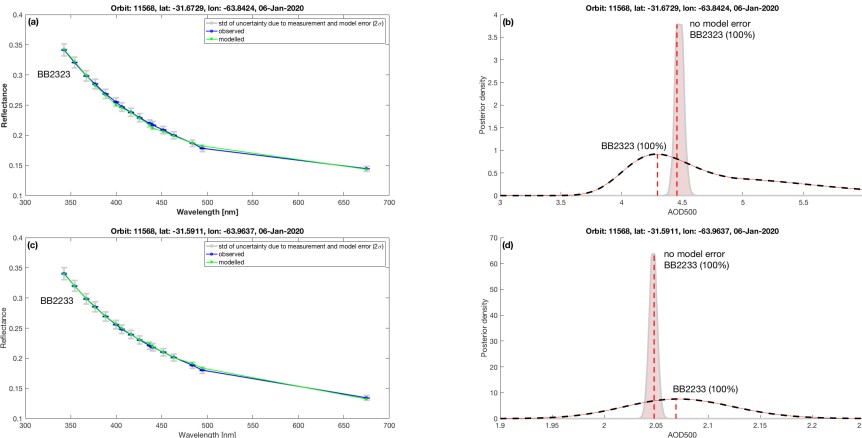

**Figure 13.** On 6 January 2020, orbit: 11568. Same as Fig. 8 but for the pixels (394,1834) collocated to AERONET site Pilar_Cordoba **(a, b)** and for the pixel nearby it (393,1836) **(c, d)**.

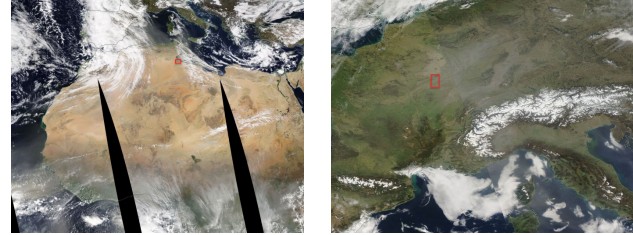

**Figure 14.** True-colour Aqua/MODIS images from the EOSDIS Worldview tool on 21 February 2021 during dust event in Northern Africa (left) and on 24 February 2021 showing dust transported to Europe (right). The areas examined are marked with red rectangles.





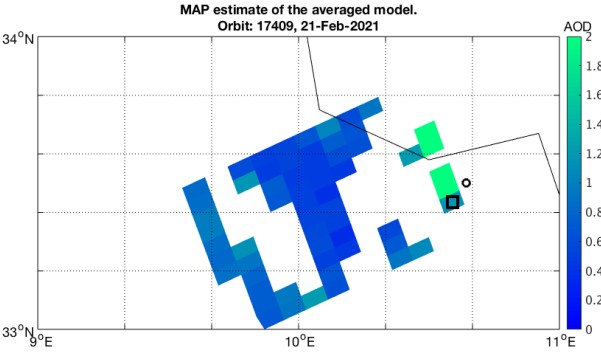

**Figure 15.** 21 February 2021, orbit: 17409. The spatial distribution of the retrieved MAP AOD estimate from the averaged posterior distribution. The location of AERONET site Medenine_IRA is marked by a black circle. Since the solution is missing for that pixel we present the more detailed results in Fig. 18 for the pixel close to it marked by a black rectangle.

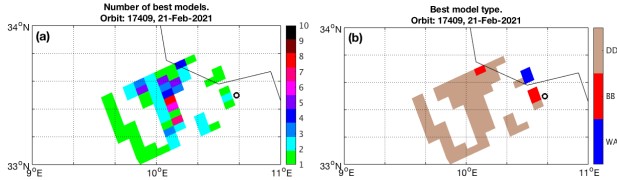

**Figure 16.** 21 February 2021, orbit: 17409. Same as Fig. 11 but for the dust case. The location of AERONET site Medenine_IRA is marked by a black circle.

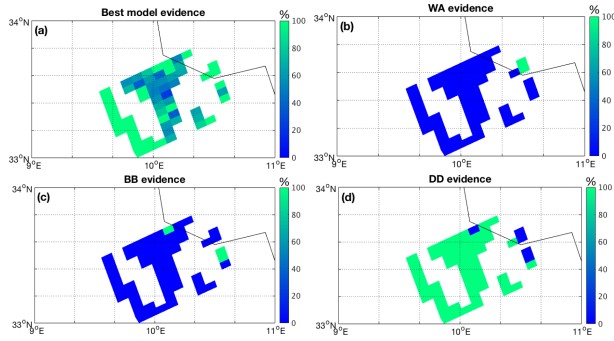

**Figure 17.** On 21 February 2021, orbit: 17409. Same as Fig. 12 but for the dust case.



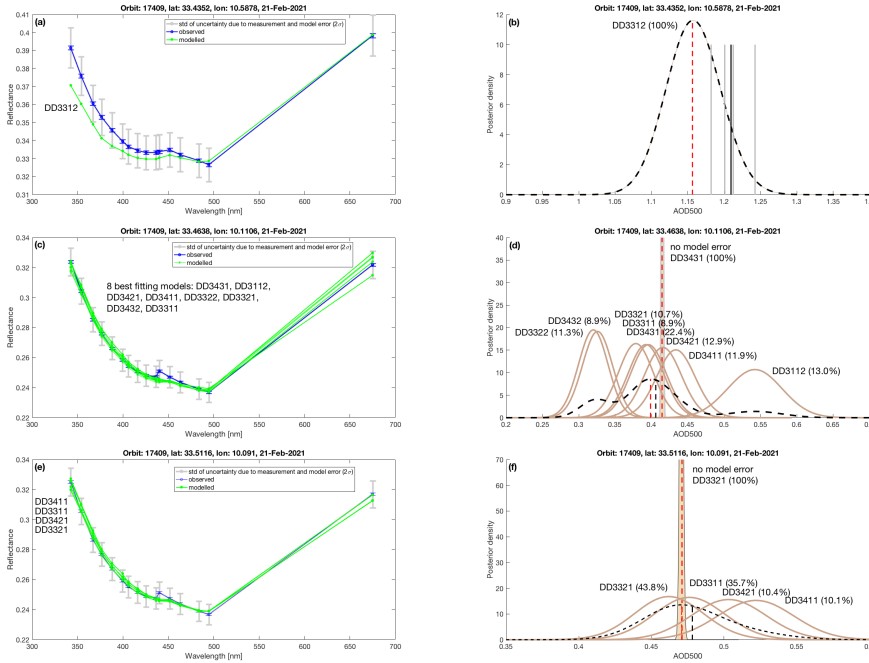

**Figure 18.** On 21 February 2021, orbit: 17409. Same as Fig. 8 but for the pixels collocated to AERONET site Medenine_IRA **(a, b)** and nearby it **(c-f)**. These TROPOMI pixels are (60,2977), (55,2981) and (55,2982) respectively. The black dashed vertical line (visible in **(d)** and **(f)**) indicates the evidence weighted sum of the MAP AOD estimates of the selected models, whereas the red dashed vertical line stands for the MAP AOD estimate from the averaged posterior distribution. The gray vertical lines in **(b)** denote the AERONET AOD(500) values within a one-hour time window including the TROPOMI overpass time and the black solid vertical line is their average value.





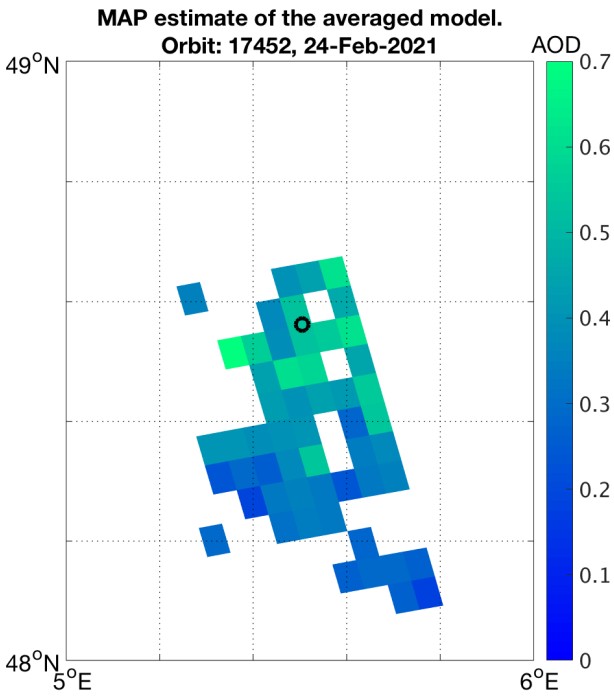

**Figure 19.** 24 February 2021, orbit: 17452. Same as Fig. 15. The location of AERONET site Bure_OPE is marked by a black circle.

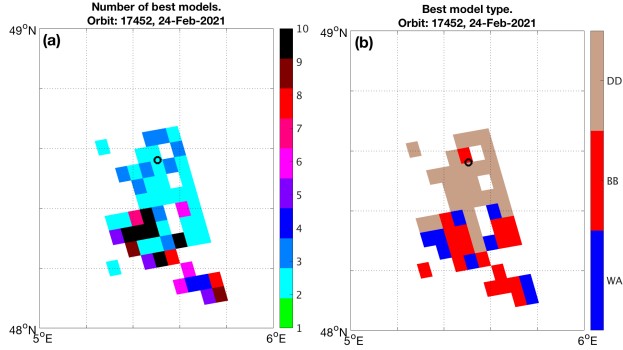

**Figure 20.** 24 February 2021, orbit: 17452. Same as Fig. 16 but showing the case when detecting dust transported to Europe. The location of AERONET site Bure_OPE is marked by a black circle.





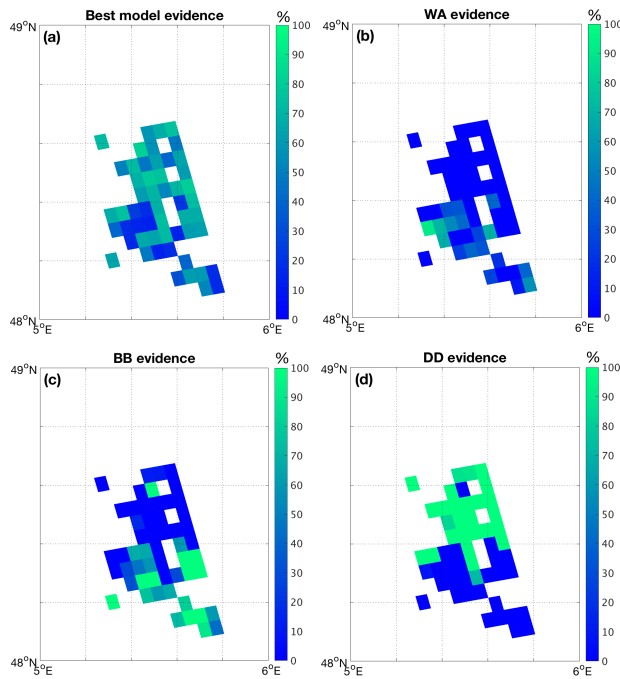

**Figure 21.** On 24 February 2021, orbit: 17452. Same as Fig. 17 but showing the case when detecting dust transported to Europe.

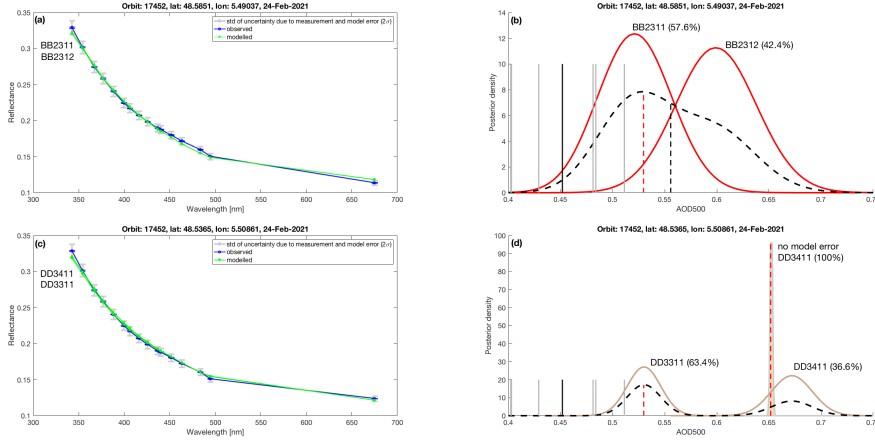

**Figure 22.** On 24 February 2021, orbit: 17452. Same as Fig. 8 but for the two pixels (274,3205) and (274,3204) collocated to AERONET site Bure_OPE. The gray vertical lines in **(b)** and **(d)** denote the AERONET AOD(500) values within a one-hour time window including the TROPOMI overpass time and the black solid vertical line is the mean value.





**Table 1.** Aerosol size distribution parameters and refractive index for weakly absorbing (WA), biomass burning (BB) and a set of desert dust (DD) type of aerosols included in LUTs. The model ID number has been coded as: [main type][imaginary part of complex refractive index][vertical distribution][size distribution]. The size distribution is given as bimodal log-normal functions. The mean particle radius, rg [$\mu$m], and the standard deviation, $\sigma$ [$\mu$m], are given for both modes, m1 and m2. In the column 'n21' is given a m2 mode fraction of the number concentration. The third digit ('x') in the model ID number of BB and DD models has a range of 1–3 indicating different vertical distributions (see Sect. 3.1). The DD type of models have wavelength dependent imaginary refractive index listed in Table 4. References: Torres et al. (2007); Veihelmann et al. (2007); Curier et al. (2008).

| Model | rg m1 | rg m2 | $\sigma$ m1 | $\sigma$ m2 | n21 | Refractive index (real) | Refractive index (imag) |
|---|---|---|---|---|---|---|---|
| WA1111 | 0.078 | 0.497 | 1.499 | 2.160 | 4.36e-4 | 1.4 | 5.0e-8 |
| WA1112 | 0.088 | 0.509 | 1.499 | 2.160 | 4.04e-4 | 1.4 | 5.0e-8 |
| WA1113 | 0.137 | 0.567 | 1.499 | 2.160 | 8.10e-4 | 1.4 | 5.0e-8 |
| WA1211 | 0.078 | 0.497 | 1.499 | 2.160 | 4.36e-4 | 1.4 | 0.004 |
| WA1212 | 0.088 | 0.509 | 1.499 | 2.160 | 4.04e-4 | 1.4 | 0.004 |
| WA1213 | 0.137 | 0.567 | 1.499 | 2.160 | 8.10e-4 | 1.4 | 0.004 |
| WA1311 | 0.078 | 0.497 | 1.499 | 2.160 | 4.36e-4 | 1.4 | 0.012 |
| WA1312 | 0.088 | 0.509 | 1.499 | 2.160 | 4.04e-4 | 1.4 | 0.012 |
| WA1313 | 0.137 | 0.567 | 1.499 | 2.160 | 8.10e-4 | 1.4 | 0.012 |
| BB21x1 | 0.074 | 0.511 | 1.537 | 2.203 | 1.70e-4 | 1.5 | 0.01 |
| BB21x2 | 0.087 | 0.567 | 1.537 | 2.203 | 2.06e-4 | 1.5 | 0.01 |
| BB21x3 | 0.124 | 0.719 | 1.537 | 2.203 | 2.94e-4 | 1.5 | 0.01 |
| BB22x1 | 0.074 | 0.511 | 1.537 | 2.203 | 1.70e-4 | 1.5 | 0.02 |
| BB22x2 | 0.087 | 0.567 | 1.537 | 2.203 | 2.06e-4 | 1.5 | 0.02 |
| BB22x3 | 0.124 | 0.719 | 1.537 | 2.203 | 2.94e-4 | 1.5 | 0.02 |
| BB23x1 | 0.074 | 0.511 | 1.537 | 2.203 | 1.70e-4 | 1.5 | 0.03 |
| BB23x2 | 0.087 | 0.567 | 1.537 | 2.203 | 2.06e-4 | 1.5 | 0.03 |
| BB23x3 | 0.124 | 0.719 | 1.537 | 2.203 | 2.94e-4 | 1.5 | 0.03 |
| DD31x1 | 0.042 | 0.670 | 1.697 | 1.806 | 4.35e-3 | 1.53 | wv dep. |
| DD31x2 | 0.052 | 0.670 | 1.697 | 1.806 | 4.35e-3 | 1.53 | wv dep. |
| DD32x1 | 0.042 | 0.670 | 1.697 | 1.806 | 4.35e-3 | 1.53 | wv dep. |
| DD32x2 | 0.052 | 0.670 | 1.697 | 1.806 | 4.35e-3 | 1.53 | wv dep. |





**Table 2.** The set of the desert dust models (DD) where the size distribution is given as the mean particle radius, rg [micron], and the standard deviation, σ [micron]. The third digit ('x') in the model ID number has a range of 1–3 and indicates different vertical distributions. Reference: Wagner et al. (2012) (Tables 1 and 5, sample Cairo 2).

| Model | rg m mean | rg m effective | σ | Refractive index (real) | Refractive index (imag) | shape | aspect ratio |
|---|---|---|---|---|---|---|---|
| DD33x1 | 0.175 | 0.27 | 1.50 | 1.53 | wv dep. | spherical | - |
| DD33x2 | 0.135 | 0.22 | 1.50 | 1.53 | wv dep. | spherical | - |
| DD34x1 | 0.175 | 0.27 | 1.50 | 1.53 | wv dep. | prolate spheroid | 0.25 |
| DD34x2 | 0.135 | 0.22 | 1.50 | 1.53 | wv dep. | prolate spheroid | 0.25 |

**Table 3.** The set of the desert dust models (DD) where the size distribution is given as the mean particle radius of accumulation mode and rough mode [micron] with geometric standard deviations espectively. Here 'n21' is the coarse mode fraction of the particle number concentration i.e. 67.1 % on average. The third digit ('x') in the model ID number has a range of 1–3 and indicates different vertical distributions. Reference: Räisänen et al. (2013).

| Model | accumulation mode | geom. stand.dev. | rough mode | geom. stand.dev. | n21 (%) | Refractive index (real) | Refractive index (imag) |
|---|---|---|---|---|---|---|---|
| DD35x1 | 0.06 | 1.59 | 1.5 | 2.0 | 67.1 | 1.52 | wv dep. |
| DD35x2 | 0.4 | 1.59 | 2.7 | 2.0 | 67.1 | 1.52 | wv dep. |





**Table 4.** The wavelength dependent imaginary part of the refractive index given for the DD models. If necessary, the imaginary refractive index values at our used wavelength bands were interpolated using the values given in the reference material.

| Wavelength | DD31xx | DD32xx | DD33xx | DD34xx | DD35xx |
|---|---|---|---|---|---|
| 342.5 | 6.06E-3 | 1.21E-2 | 1.33000E-2 | 1.33000E-2 | 1.6666667E-2 |
| 354.0 | 5.61E-3 | 1.12E-2 | 1.25640E-2 | 1.25640E-2 | 1.2900000E-2 |
| 367.0 | 5.02E-3 | 1.00E-2 | 1.16360E-2 | 1.16360E-2 | 7.7000000E-3 |
| 376.5 | 4.55E-3 | 9.10E-3 | 1.09520E-2 | 1.09520E-2 | 3.9000000E-3 |
| 388.0 | 4.05E-3 | 8.10E-3 | 1.01240E-2 | 1.01240E-2 | 2.3000000E-3 |
| 399.5 | 3.62E-3 | 7.24E-3 | 9.29600E-3 | 9.29600E-3 | 2.0125000E-3 |
| 406.0 | 3.40E-3 | 6.80E-3 | 8.84200E-3 | 8.84200E-3 | 1.8500000E-3 |
| 416.0 | 3.10E-3 | 6.20E-3 | 8.26200E-3 | 8.26200E-3 | 1.6000000E-3 |
| 425.5 | 2.84E-3 | 5.68E-3 | 7.71100E-3 | 7.71100E-3 | 1.4725000E-3 |
| 436.5 | 2.58E-3 | 5.16E-3 | 7.07300E-3 | 7.07300E-3 | 1.4175000E-3 |
| 440.0 | 2.51E-3 | 5.02E-3 | 6.87000E-3 | 6.87000E-3 | 1.4000000E-3 |
| 451.5 | 2.31E-3 | 4.62E-3 | 6.20300E-3 | 6.20300E-3 | 1.3425000E-3 |
| 463.0 | 2.14E-3 | 4.28E-3 | 5.80800E-3 | 5.80800E-3 | 1.2925000E-3 |
| 483.5 | 1.88E-3 | 3.76E-3 | 5.31600E-3 | 5.31600E-3 | 1.2412500E-3 |
| 494.5 | 1.76E-3 | 3.52E-3 | 5.05200E-3 | 5.05200E-3 | 1.2137500E-3 |
| 500.0 | 1.69E-3 | 3.38E-3 | 4.92000E-3 | 4.92000E-3 | 1.2000000E-3 |
| 675.0 | 1.69E-3 | 3.38E-3 | 2.26000E-3 | 2.26000E-3 | 9.8181818E-4 |
| Reference: | Veihelmann et al. (2007) | Veihelmann et al. (2007) | Wagner et al. (2012) | Wagner et al. (2012) | Räisänen et al. (2013) |

**Table 5.** Node point entries of the multi-dimensional LUTs.

| Parameter | Symbol | N of entries | Entries |
|---|---|---|---|
| Wavelength | $\lambda$ | 19 | 331.7, 340.0, 342.5, 354.0, 367.0, 376.5, 388.0, 399.5, 406.0, 416.0, 425.5, 436.5, 440.0, 451.5, 463.0, 483.5, 494.5, 500.0, 675.0 |
| Surface pressure | $p_s$ | 2 | 554 [hPa], 1013 [hPa] |
| Cosine of Solar Zenith Angle | $\mu_0$ | 8 | 1.0, 0.9, 0.8, 0.7, 0.6, 0.5, 0.4, 0.3 |
| Cosine of Viewing Zenith Angle | $\mu$ | 8 | 1.0, 0.9, 0.8, 0.7, 0.6, 0.5, 0.4, 0.3 |
| Relative azimuth angle | $\Delta\phi$ | 19 | 0, 10, 20, 30, 40, 50, 60, 70, 80, 90, 100 110, 120, 130, 140, 150, 160, 170, 180, |
| Aerosol optical depth at 500 nm | $\tau$ | 9 | 0, 0.1, 0.25, 0.5, 1.0, 1.5, 2.5, 5.0, 10.0 |





**Table 6.** The variables in the LUT resulting from the RT simulations. These variables are used for to calculate the modelled TOA reflectance in Eq. (2).

| Variable name | Symbol | Dimensions | Dependencies |
|---|---|---|---|
| Atmospheric path reflectance | $R_a$ | 6D | $(\lambda, \tau, \mu, \mu_0, \Delta\phi, p_s)$ |
| Spherical Albedo | $s$ | (19, 9, 2) | $(\lambda, \tau, p_s)$ |
| Transmittance | $T$ | 5D | $(\lambda, \tau, \mu, \mu_0, p_s)$ |

**Table 7.** The collocated AERONET sites and the results of the case studies. The AERONET AOD(500) is the average of the AOD at 500 nm values within a one-hour time window coinciding with the TROPOMI overpass time. The TROPOMI AOD is the MAP AOD estimate from the averaged posterior density. We have used the AERONET Version 3 direct-sun Level 2 or Level 1.5 data as reference. The results are reported for the TROPOMI pixels collocated or nearby (marked by [*]) to the AERONET sites.

| Date | Site name AERONET | (Lat,Lon) AERONET | AOD(500) AERONET | AOD Level AERONET | Orbit TROPOMI | Pixel ind TROPOMI | AOD TROPOMI |
|---|---|---|---|---|---|---|---|
| 23 Aug 2020 | NASA_Ames | (37.4° N, 122.1° W) | 2.129 | Level 2 | 14832 | (50,2605)[*] | 2.29 |
| 23 Aug 2020 | Monterey | (36.6° N, 121.8° W) | 1.575[a] | Level 1.5 | 14832 | (47,2589) | 3.03 |
| 23 Aug 2020 | Univ_of_Nevada-Reno | (39.5° N, 119.8° W) | 0.672 | Level 2 | 14832 | (84,2627) | 0.65 |
| 6 Jan 2020 | Pilar_Cordoba | (31.7° S, 63.9° W) | 0.719 | Level 1.5 | 11568 | (394,1834) | 4.29 |
| 21 Feb 2021 | Medenine_IRA | (33.5° N, 10.6° E) | 1.209[b] | Level 1.5 | 17409 | (60,2977)[*] | 1.16 |
| 24 Feb 2021 | Bure_OPE | (48.6° N, 5.5° E) | 0.452 | Level 1.5 | 17452 | (274,3205) | 0.53 |
| 24 Feb 2021 | Bure_OPE | (48.6° N, 5.5° E) | 0.452 | Level 1.5 | 17452 | (274,3204) | 0.53 |

[a] including two measurements within two hours before TROPOMI overpass time [b] interpolated using Ångström exponent 440-675 nm