# Peer review of "Bayesian uncertainty quantification in aerosol optical depth retrieval applied to TROPOMI measurements"

_Atmospheric Measurement Techniques, 2021_

## Author Comment (AC1)

The authors would like to thank the Reviewer for time and effort used to evaluate the manuscript. The comments are valuable and help to improve the manuscript. We have taken the comments into account in the revised manuscript. In addition we have improved the use of English as suggested by the reviewer, and corrected some typos.
The detailed replies (black font) to all the reviewer comments (blue font) are given below.
The pages, line numbers and equation numbers refer to the manuscript under discussion.

**Referee #1**

The paper discusses a Bayasian approach of model selection in satellite AOD retrievals. Aerosol model selection in single-view satellite aerosol retrievals is the largest source of errors, and a method to better control the selection of different aerosol optical properties would be very benificial for AOD retrievals using spectrometers and imagers. In this paper the spectral information is used to select aerosol models and estimate the goodness-of-fit. Instead of one aerosol model, a mix of models can be found which can better fit the observations with noise.

The subject of the paper is interesting, however, there is a lack of clear new information about the approach, the applicability and results. The paper extensively describes the AOT retrieval approach and established procedures, but lacks in the escription of the new method and results. Up to section 3, the paper is well structured and reads like an AOD retrieval paper, but from section 4, the description becomes unclear and limited. Several times, the authors refer to the method like in OMI, without signifying what the benefit is for new instruments, like TROPOMI in this case. Since only a few cases are presented, which show often the limitation rather than the benefit of the method, the use for TROPOMI or other instruments is difficult to establish.

**Reply:** We introduce here Bayesian retrieval approach that can be implemented to TROPOMI AOD retrieval. The algorithm takes into account forward model uncertainty as well as includes model selection capability. Similar Bayesian model selection algorithm was first introduced to OMI AOD retrieval by Määttä et al, 2014. Here we expand similar methodology to the next generation satellite instrument TROPOMI, which has high spatial resolution, low measurement noise and wide wavelength coverage. The extended wavelength range to NIR is important enhancement as the longer wavelengths may carry additional information the AOD retrieval can take advantage of. With the improved signal-to-noise ratio the relative importance of the modelling uncertainty increases which is therefore important to take into account in the retrieval. This is a research type of retrieval and aim is to study the Bayesian approach for uncertainty quantification as well as consider model selection problem. As we stated in the Abstract (p1 14-16) we see that this kind of method can be used to experiment with different aerosol types and evaluate the most probable aerosol types by the model selection method. This can be considered as a benefit of this method.

Here, the focus is to study how the methodology can be expanded to TROPOMI and experiment the applicability of the approach when the setup has changed including noise level, LUTs, wavelength range and surface reflectance data. The possibility to generate LUT models for particular aerosol properties benefits examination the capability of the method to select proper aerosol models, as well as it can bring new information about the model selection process. By including dust models with non-spherical shape represent new type of models in this context.

We have not done comparison study of the methodology applied to TROPOMI data versus using OMI data. We agree that such comparison would be interesting as a further study since the TROPOMI has improved observations with higher spatial resolution than the OMI has.

We agree with the reviewer that the presented case studies and limited results are not adequate to confirm correct functioning of the methodology more widely. While it is shown here that the methodology is very promising, using the algorithm e.g. for satellite AOD comparison, one would require more comprehensive testing and expanding the LUTs, such work is considered to be outside the scope of this paper.

These issues are also discussed in our responses to the specific comments P15 l558 and P16 478. The added value of the study presented in this paper is discussed as a response to the specific comment P9 l253.

One thing is at least unclear: The retrieval of AOD may improve from using a statistical mix of aerosol models, over selecting only one model. However, would the retrieved SSA and Angstrom exponent also be determined from the mix? I did not find this in the paper, while especially the derived microphysical properties are derived from the chosen aerosol model, and hence especially sensitive to changes.

**Reply:** This is an interesting question. The presented methodology have been applied here only for retrieving AOD. We have not studied its capability to make inference about SSA and Ångström exponent from the mixture of selected LUT models.
Our response to the specific comment P11 l305 handle this question in more detail.

The paper lacks a clear comparison of cases with and without the new approach. Although the selected cases do compare the new approach with the aproach without model error, the cases are often extreme cases, that fail to show reasonable results and the difference between the new and old approach is difficult to establish. Moreover and more importantly, the new approach is given as is (and not too well described) and it is unclear what the various possibilities and uncertainties of the approach are. A sensitivity study with varying parameters may help to show the limitations of the new approach.

**Reply:** We would like to clarify that we present only one approach and that takes into account forward modelling error (i.e. model discrepancy). For the sake of comparison we show also results when the model error was not included in order to see the difference in the solution i.e. in the retrieved posterior distribution that also signifies the retrieved AOD uncertainty. When the model error is not taken into account the estimated uncertainty is more optimistic, i.e. the width of the posterior density is more narrow.

As the reviewer suggested, to study the sensitivity of the method to the sources of uncertainty e.g. to the surface reflectance or to the noise level (measurement bias and model error) would be important, but it would need more detailed study so it can be subject for the further study.

**Changes to manuscript**: For clarity we have stated in the revised manuscript that the results without model error are shown only for interest in order to illustrate the difference in the measure of uncertainty for the retrieved AOD (p12).

The use of English should be improved.

**Reply:** We have proofread and improved the use of English in the revised manuscript.

**Specific comments:**

P8l231-l238. The description of the Gaussian variorum model is not comprehensive and difficult to understand. It's not properly explained how the values of the parameters λ,σ0 and σ1 are derived.  However, this is the core of the paper: the correlation found by the statistical spectral dependence between the aerosol models determines the added value of the Bayesian uncertainty quantification. It seems to me that the way the approach is set up is essential for the study results. A proper description of the derivation of these essential parameters seems to be the least to demand, and a sensitivity study of the results based on various settings would also be proper.

**Reply:** As suggested by the reviewer, we will give more comprehensive description about how the values for the parameters l, $\sigma_0^2$ and $\sigma_1^2$ are derived. The reviewer has right that these parameter values are essential since they are used to characterise the model discrepancy (i.e. the forward model error). The parameter values are derived using the empirical semivariogram and then fitted the parametric Gaussian semivariogram model (this formula will be added as well). We will add a figure (see Figure (left) below) and reword the text accordingly.
The sensitivity of the results to the different parameter values used for model error covariance matrix will be illustrated by an another figure added (see Figure (right) below).

**Changes to manuscript:** We have added a figure (see Figure below) showing (left) the values of the empirical semivariogram and the fitted Gaussian variogram model with different parameter

values for $\sigma_0^2$, $\sigma_1^2$ and l. (right) Changes in the posterior probability distribution when the parameters correlation length l and $\sigma_1^2$ were altered.
We have added a formula for the theoretical Gaussian semivariogram model and reworded the text part (Page 8 lines 231-240) as:

"Next, we fitted a theoretical parametric Gaussian semivariogram model from the literature (Banerjee et al., 2004)

$$\gamma(d) = \begin{cases} \sigma_0^2 + \sigma_1^2[1 - \exp(-(\frac{d^2}{l^2}))], & \text{if } d > 0 \\ 0, & \text{otherwise} \end{cases}$$

where $d=|\lambda i - \lambda j|$, to the empirical semivariogram in order to find the values for tuning parameters l, $\sigma_0^2$ and $\sigma_1^2$ of the theoretical model (see Fig. X left). A correlation length l describes the wavelength distance where the residuals are still correlated. In addition, a parameter $\sigma_0^2$ is responsible for non-spectral diagonal variance and a parameter $\sigma_1^2$ for spectral variance. We have chosen the parameter values as l=90, $\sigma_0^2$ =1.0e-6 , and $\sigma_1^2$ = 1.0e-4 (see Fig. X (left), black curve). Following the Gaussian variogram model we derive the covariance function C of the Gaussian process model as
<Eq. (4)>
It depends on the wavelength distance and determines the correlation properties of the model discrepancy. Finally, the covariance function forms the corresponding model error covariance matrix C that defines the allowed smooth departure for the modelled reflectance from the observed reflectance. For computational issues we ended up to use the parameter values as l=90, and for both $\sigma_0$ and $\sigma_1$ values of 1% of the measured reflectance when the model error covariance matrix C was implemented.
In Fig. X (right) is illustrated the sensitivity of the resulting posterior probability distribution of AOD to the different values for the covariance function parameters l and $\sigma_1^2$. We can observe that the posterior reflects the change in the l and $\sigma_1^2$. For the sake of comparison it is also plotted the posterior density if the model error was not taken into account"

[Figure]

[Figure]

**Figure.** (**Left**) The Gaussian parametric semivariogram model (solid curves) with different values for tuning parameters $\sigma_0^2$, $\sigma_1^2$ and correlation length l is fitted to the empirical semivariogram (circles). (**Right**) The resulting posterior probability distributions corresponding to the different parameter values for $\sigma_1^2$ and l, and when the model discrepancy (MD) was not included.

P9l246: " ..the ratio of successful retrieval was ~ 39 %." It is not clear what ratio is referred to here, it is not described. However, I suspect from the next sentences that it is something like the number of aerosol models that deliver a valid retrieval within the noise range. Again, this is essential for the method that is the subject of this paper and should be much more elaborated on.

**Reply:** We agree, this sentence is unclear and we will revise the sentence to be more understandable in the revised manuscript.

**Changes to manuscript**: We reworded this sentence as:
"The number of pixels processed was 2162 for 24 July and 617 for 22 March respectively.

The ratio of the number of successfully retrieved pixels to the number of pixels processed was ~39 % for 24 July and ~12 % for 22 March respectively."

P9 l253: "We will present only the main idea here as the methodology based on the Bayesian inference is explained thoroughly in the papers (Määttä et al., 2014; Kauppi et al., 2017) when applied to the OMI measurements. " If only a few sets of TROPOMI measurements were processed, and no new methodology is introduced, what is the added value of this paper?

**Reply:** We thank the reviewer for this remark. We need to be more explicit about the added value of this paper.

This paper presents the setup and some results when the methodology has now been applied to the TROPOMI data. In the former papers (Määttä et al., 2014; Kauppi et al., 2017) this methodology was introduced and first time applied to satellite data using OMI measurements. What is new in this paper are
1) we have expanded the wavelength range by including 675 nm
2) for this study we have created the aerosol model LUTs for the TROPOMI measuring geometry and for the selected wavelengths by radiative transfer simulations, as we didn't have these LUTs at hand
3) we have included dust models with particles of non-spherical shape representing new type of models in this context
4) for this application with TROPOMI data we have re-estimate the model discrepancy (i.e. forward model error) using the TROPOMI measured reflectance and the created LUT models.
In the Introduction section the items 1), 2) and 3) above are already mentioned (p3 57-58, p3 65-69), but we will also include item 4) in the same chapter in the revised version.

As a conclusion, the added value of this paper is to show that by changing the setup (e.g. noise level, LUTs, wavelength range and surface reflectance data) the methodology can be applied to different satellite instruments. In addition, one focus in this study was to confirm that the Bayesian model selection based methodology enables to experiment with different aerosol types and evaluate the most probable aerosol model selection from LUTs constructed for the desired aerosol properties.

**Changes to manuscript:** We have added text regarding item 4) in the Introduction section. We have rewritten the first part in the Discussion and Conclusions section (p 14) and emphasized the added value of this study.

P10 l302: The difference between the MAP AOD estimate and the weighted sum MAP AOD is difficult to determine from the text and the figures. The red line is the mode from the averaged posterior distribution. However, the black line is the sum of the weighted MAP estimates of the individual models (I assume the same (number) of models that make up the averaged posterior distribution. Then how does the difference come about? By the different order of averaging? Or something completely else that I might be missing? What is the significance of the two?

**Reply:** We thank the reviewer for pointing out the confusion caused by the two separate point AOD estimates shown in the plots. This is the case when there are more than one model selected as the best model to explain the measurement.

In short, the difference is that for the MAP AOD estimate from the averaged posterior density (Eq. 8) the sum is taking over the posterior densities weighted by the models' relative evidences, i.e. we first combine the posterior distributions of the selected models and after that determine the point estimate for AOD. Whereas, the weighted mean of the MAP AOD estimates (formerly called as sum of the MAP estimates in the text) is taking over the AOD values that are the MAP estimates from the individual models (i.e. MAP estimates from the separate posterior distributions).
As the reviewer assumed, the number of selected models are the same, as well as the weights (i.e. the relative evidences) are the same.

The primary point estimate for AOD is the MAP estimate from the averaged posterior density (red dashed vertical line). The significance of using Bayesian model averaging technique (Eq. 8) is that

it accounts for the uncertainty in model selection, and in the form of averaged poster curve provides the shared inference about the range of AOD values.

In order to avoid confusion we will rename the alternative point measure for AOD (indicated by black dashed vertical line) as "the weighted mean of the MAP AOD estimates".

The difference between these two separate point measures for AOD, when resulting from the mixture of the LUT models, can be seen in the Fig.2 shown below.

[Figure]

Figure 2., page 21 in the manuscript

The relative evidence (%) (i.e. the weight $w_i$) is reported next to each model ID number
**Left panel**: The MAP AOD estimate from the averaged posterior density (dashed red vertical line) and the weighted mean of the MAP AOD estimates (dashed black vertical line) are about the same.
**Middle panel**: The two AOD point estimates differ. As seen the averaged posterior has two peaks, but since the model WA1213 is superior to the other two models it produces a higher peak. Whereas, the weighted mean of the MAP AOD estimates is determined independently from the averaged posterior curve, just calculated as $w_i*AOD_i$ where $AOD_i$ is the MAP estimate from the posterior within the model $m_i$.

It is worth of note that when the averaged posterior distribution has more than one peak it indicates difficulty in model selection i.e. there are distinct potential solutions. Evidently it signifies that there is a mixture of different aerosol types.

**Changes to manuscript**: We reworded the designation for "the weighted mean of the MAP AOD estimates" in the text where it appears (p10 280-282, p10 300-303, and p14 431). We revised the figure text for Fig. 2 and Fig.18 accordingly.
We have also added discussion about the difficulty in model selection when the averaged posterior has more than one peak, referred to Fig.2. (p10).

P11: l305: I understand the AOT is some kind of weighted average from the selected best models. However, these best models can be very different types, WA and BB. Are derived aerosol property retrievals like SSA and Angstrom exponent also derived from the same mix of models?

**Reply:** It would have been possible to estimate SSA and Ångstrom exponent from the mixture of the selected aerosol models. But it needs further examination how to take advantage of relative evidence values for the selected models when, for instance, retrieving total SSA by weighting SSA of each aerosol model with its relative AOD-weight (i.e. extinction weight). However, we believe that these SSA estimates would have included significant uncertainty, since in this way we would not have specifically used the information from the shortest UV channels, which are important to infer aerosol absorption (e.g., as in OMAERUV algorithm to retrieve SSA). Moreover, it is to be emphasized that we deliberately focused on AOD retrievals, where our methodology clearly has its strengths, thus SSA or Ångström exponent retrievals were outside the scope of our study.

**Changes to manuscript**: We have added the following note in the revised manuscript, in the beginning of the section Discussion and Conclusions: "We like to note that we have not examined in this work estimation of SSA and Ångström exponent from the mixture of the selected models." In order to avoid misunderstanding, we have added a link for the AERONET data (https://aeronet.gsfc.nasa.gov) when referring Ångström exponent instead of writing "(not shown here)" (p12 359, p13 387 and p14 411).

P13: The retrieval around Pilar Cordoba of AOD > 4 where the AERONET retrieval is only 0.7 seems unlikely, especially so far from the source, although the high UV aerosol index does indicate something highly absorbing here. It is a clear failure of the satellite retrieval. This is particular true for the non-model error approach, which retrieves generally, and also now, higher values. However, I don't understand why in this case only one aerosol model is selected in the Bayesian approach, resulting in a high AOT retrieval. The authors state that the range of AOT is large, with much lower AOT 'a little bit further', which is not very satisfactory, and that the range of BB models is insufficient. This is a clear lack of the approach then, when the strength should be to include the differences between many different models. Would an easy solution not be to include a few aerosol model with extreme values outside the current aerosol properties, instead of using 66 models with only very small differences between them?

**Reply:** We agree with the reviewer that in the Pilar Cordoba case the retrieval does not give satisfactory solution. However, the reflectance from the selected best model has a good fit to the measurement as can be seen in Fig, 13 (left panel), that makes this case interesting, and was the reason to include this example case in the manuscript. This gives rise to consider if it is possible to generate correct aerosol LUT model(s) for this case.
As the reviewer suggested, it would be interesting to include LUT models with extreme aerosol properties. Yet, it requires effort to collect correct input (i.e. aerosol properties) for RT calculations in order to generate LUTs.

As the reviewer pointed out in this Pilar Cordoba case with retrieved high AOD level only one LUT model was selected. In practise, there are less suitable LUT models that fit to the measured reflectance when the corresponding AOD level gets higher.
For instance, see Figure below (related to our response to the specific comment P15 l457), where is shown in the rightmost panel for the two distinct aerosol models the spectral AOD values for nine different AOD levels stored in LUTs. As can be seen the shape of the spectral AOD of these models differentiate more and more when AOD level gets higher. The AOD levels at 500 nm are the same for the separate models since the values at node points in LUTs are fixed (see Table 5 in p34).

**Changes to manuscript**: None, since we discuss about the possibility of lack of proper aerosol models in the manuscript already (e.g. p13 386, p15 459 and p16 475).

P15 l457: "As a special feature in this study we have included aerosol models of dust type with non-spherical shape of particles. The particle shape can have a large effect on the scattering properties. " Unfortunately, this is not further investigated in the paper, it is only noted in the paper that some non-spherical models are sometimes selected.

**Reply:**
We have now added text parts and a figure (see below) that discuss the effect of particle shape to the scattering properties. This was investigated for the two LUT models DD3311 and DD3411 which differentiate only by the particle shape (Table 2 and Table 4.). These models were also selected as a result for the case study in Sect. 5.3 and shown in Fig. 22 (c, d).

**Changes to manuscript**:

We added the following figure (numbered as Fig. 2) in the revised manuscript:
We added the following text part to the section 3.1.2:

"As for example, the two dust models, DD3311 and DD3411, differ solely by the shape of the dust model particle. The implications to the key optical properties are depicted in Fig. 2. The angular

[Figure]

**Figure**. Differences in the optical properties of spherical (black lines, DD3311) and non-spherical particles (grey lines, DD3411). (a) The angular distribution of the intensity of scattered light (S11) is shown with solid lines for UV and dashed lines for the largest wavelength considered. (b) SSA shows the relative strength of scattering to total extinction as a function of wavelength. (c) The wavelength dependence of the Ångstrom exponent for both particle shapes. (d) AOD from the RT computations using these two particle models.

distribution of scattered intensity S11 shows that the shape has a larger impact on S11 for smaller wavelengths, i.e., when the particles are relatively larger as compared to the wavelength. The spheroidal particles are more forward-scattering while the spherical particles can have up to 5 times stronger backscattered intensity. This difference decreases significantly for larger wavelengths. Single-scattering albedo (SSA) shows that the non-spherical particles scatter light more than the spherical particles for wavelengths smaller than 650 nm. For larger wavelengths, the spherical particles scatter more light than the non-spherical particles, which is also seen to have an impact on AOD. The Ångstrom exponent shows a large difference between the two shapes despite the equal volumes of the individual particles. Due to the shape difference, the physical sizes of the particles differ, which is reflected in the resulting Ångstrom exponent values."

We added the following text part to the section 5.3:

"The difference in the retrieved AOD between these two shapes originates from the differences in the optical properties of the models. As pointed out by the comparison (see Fig. 2), the differences are particularly large at smaller wavelengths (in the UV) and concern specifically the angular distribution of scattered intensity and the fraction of scattered light compared to extinction. In particular the former will result in different estimates depending on the geometry of the satellite retrieval (i.e. the SZA and VZA)."

In addition, we reworded the sentence in p15 457 by adding reference to the Sect. 3.1.2.

P15 l558: "It is expected that the aerosol properties included do not cover all the possible aerosol scenarios." I agree to this statement. I think this would be worthwhile to investigate further and use the Bayesian inference to use it properly.

**Reply:** (p15 458) We thank the reviewer for those words of encouragement to use the presented methodology for investigate further with different aerosol scenarios.
This is mentioned in the section Discussion and Conclusions where is a list of suggestions for further studies (items 1 and 2. p16).

P16 478: "We need to do more retrieval exercises and verify the results e.g. with the ground-based AERONET data before we can make conclusions about the retrieval accuracy. " I agree. In the paper a few case were selected to study the effects of different models, and understandably extreme cases of smoke and dust were selected. However, it seems that especially these cases are not well represented by the aerosol models. It might make sense to select more moderate

cases, that make up the majority of satellite retrievals, and see how the specific new approach of using Bayesian interference improves the majority of the cases.

**Reply:** We agree, the case studies presented represent extreme cases. A dedicated study for experimenting the methodology with comprehensive selection of different aerosol scenarios including moderate cases is needed.

**Changes to manuscript**: We have reworded the sentence as "We need to do more retrieval exercises with a comprehensive selection of different aerosol scenarios including moderate cases and verify the results e.g. with the ground-based AERONET data before we can make conclusions about the retrieval accuracy."

---

## Author Comment (AC2)

The authors would like to thank the Reviewer for evaluation the manuscript and the helpful comments that improve the manuscript. We have taken the comments into account in the revised manuscript.
The detailed replies (black font) to all the reviewer comments (blue font) are given below.
The pages, line numbers and equation numbers refer to the manuscript under discussion.

**Referee #2**

**General comments**
The manuscript is devoted to a very important problem: correct model selection taken into accounting the uncertainties due to forward model approximations. The results are demonstrated and analyzed on several cases. The application of the developed method to different satellite remote sensing applications is described in the manuscript.

Overall, the manuscript is well written. Presented method and results can be interesting for broad remote sensing scientific community.

**Reply:** We like to thank the reviewer for the positive and encouraging comment.

**Specific comments**
The manuscript presents the method accounting for uncertainties due to forward model approximations. To model top of atmosphere measurements the approximation of RT based on assumption of Lambertian surface reflectance is used. This RT-approximation may introduce additional uncertainties in comparison to the case when full surface BRDF is taken into account together with correct accounting for surface and atmosphere coupling. These uncertainties depend on the observation geometry, in particularly, on the solar and observation zenith angles. What is important for these studies, they also depend on different combination of surface and aerosol properties as well as on aerosol optical depth and may affect the selection of best model. Some discussion of this problem would be interesting to see in this manuscript.

**Reply:** We thank the reviewer for bringing up this important issue related to the uncertainty due to incorrect surface reflectance assumption and not correctly accounting for radiative coupling between the atmosphere and surface.

The effect of surface reflectance assumptions to the forward model error has not been studied in more detail in this work, but the intention is to use as correct surface reflectance data as possible, e.g. full surface BRDF, in the further studies.
Since we have empirically estimated the forward model error using the residuals of model fits, i.e. observedR-modelledR, it could be possible to analyse the different combined effect of surface reflectance and aerosol properties to the forward model error, and its influence to the best model selection.

We will remove a following sentence in p7 line 212: "We assumed the Lambertian surface when simulating the LUT's reflectances with RT model.". It is misleading and unrelevant here since the Eq. (2) reveals that the surface reflectance is not accounted for until when computing modelled TOA reflectance.

**Changes to manuscript**:
We have removed the sentences in p7 lines 212-213 as misleading information.
As suggested by the reviewer we have added the following paragraph to the Section Discussion and Conclusion:
"The difficulty in the satellite aerosol retrieval is how to take into account the surface reflectance as well as surface and atmosphere coupling correctly. These sources of uncertainties in the aerosol retrieval are dependent on the observation geometry, i.e. the solar and observation zenith angles. The effect of improper surface reflectance assumption on the forward model uncertainty has not been considered separately in this paper. However, the surface reflectance is implicitly included in the forward model error as it is empirically estimated based on the residuals of model fits. If using the direction dependent surface reflectance assumptions, i.e. full surface bidirectional reflectance distribution function (BRDF), together with the correct coupling of surface and atmosphere it may affect the forward model uncertainty. The proposed methodology enables to

analyse the different combined effect of surface reflectance and aerosol properties to the forward model error, and its influence to the best model selection."